# Shift-Dependent Asymmetry: Orthogonal Inverse Low-Rank Adaptation for Federated Medical Segmentation

Xingyue Zhao [* 1 2]   Wenke Huang [* 3]   Linghao Zhuang [* 4 2]   Haoran Wu [2]   Anwen Jiang [5]   Zhifeng Wang [6]
Wenwen He [7]   Ming Feng [1]   Mang Ye [7]   Bo Xu [2]

## Abstract

Low-Rank Adaptation (LoRA) enables efficient federated fine-tuning of segmentation foundation models for medical imaging. However, most federated LoRA methods adopt a uniform aggregation rule, which breaks under the encoder–decoder asymmetry in medical segmentation: the encoder is dominated by appearance shifts, while the decoder is dominated by supervision variations. This mismatch entangles shared anatomy with site-specific biases and harms generalization. To address this, we propose Inverse Asymmetric Tuning (IAT). IAT aligns adaptation with heterogeneity sources by personalizing module-specific components in the encoder to absorb appearance shifts and in the decoder to accommodate site-dependent supervision, while retaining a shared pathway for transferable consensus. However, structural separation alone is insufficient under LoRA's bilinear parameterization, where multiplicative coupling can still cause site-specific updates to leak into the shared direction. We therefore introduce a Subspace Orthogonality Regularizer that penalizes shared–local collinearity in the effective update space, mitigating leakage without extra communication. Experiments show consistent improvements over strong federated LoRA and parameter-efficient FL baselines.

---

[*]Equal contribution   [1]Department of Neurosurgery, Peking Union Medical College Hospital, Chinese Academy of Medical Sciences and Peking Union Medical College [2]The Key Laboratory of Cognition and Decision Intelligence for Complex Systems, Institute of Automation, Chinese Academy of Sciences [3]College of Computing and Data Science, Nanyang Technological University, Singapore [4]Institute of Basic Medical Sciences, Chinese Academy of Medical Sciences and Peking Union Medical College, Beijing, China [5]Xinjiang University [6]Ant Group, China [7]Wuhan University. Correspondence to: Haoran Wu <wuhaoran2018@ia.ac.cn>, Ming Feng <fengming@pumch.cn>, Bo Xu <xubo@ia.ac.cn>.

*Proceedings of the 43rd International Conference on Machine Learning*, Seoul, South Korea. PMLR 306, 2026. Copyright 2026 by the author(s).

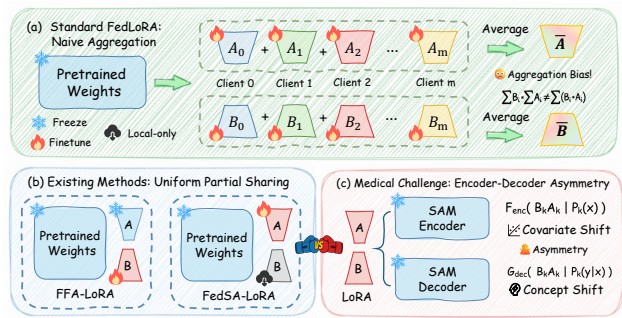

*Figure 1.* **Illustration of the motivation.** (a) **Standard FedLoRA** employs naive aggregation of all parameters, suffering from high communication costs and severe client drift. (b) **Existing Methods** (e.g., uniform partial sharing) enforce a static splitting rule across the entire network, overlooking the distinct roles of different modules. (c) **Medical Challenge: Encoder-Decoder Asymmetry.** We identify a critical structural misalignment: the encoder is primarily dominated by *covariate shift* (appearance variations), while the decoder is dominated by *concept shift* (supervision targets).

## 1. Introduction

Medical image segmentation is fundamental to clinical decision-making, ranging from organ delineation to tumor assessment (Cao et al., 2023; Hu et al., 2025c; Zhu et al., 2026). Developing robust segmentation models typically requires diverse data from multiple centers (Hu et al., 2025a; Zhu et al., 2025); however, aggregating patient images is often restricted by strict privacy regulations and data governance policies (Jiang et al., 2022). Federated Learning (FL) (McMahan et al., 2017; Huang et al., 2026) addresses this challenge by enabling collaborative training across institutions without exchanging raw data. However, the inherent data heterogeneity across sites often limits the performance of conventional lightweight models. To address this, there is a growing trend to deploy Segmentation Foundation Models (e.g., SAM) (Kirillov et al., 2023), as their robust pre-trained representations offer superior generalization against domain shifts (Liu et al., 2024; Zhang et al., 2026).

Yet, fine-tuning such massive backbones in federated settings imposes prohibitive communication and computational burdens on local computing resources. To make collaborative training feasible, Parameter-Efficient Fine-Tuning (PEFT) has become indispensable. In particular, Low-Rank

Adaptation (LoRA) (Hu et al., 2022) has emerged as the dominant strategy for federated learning (Yi et al., 2023; Zhang et al., 2024; Liao et al., 2025), as it significantly reduces transmission overhead by updating only lightweight low-rank matrices while freezing the pre-trained backbone. Accordingly, we center our investigation on the Federated LoRA paradigm.

However, standard aggregation of LoRA weights suffers from an inherent inconsistency due to its bilinear parameterization ($\Delta W = BA$). Since matrix multiplication is non-linear, averaging decomposed factors on the server generally fails to reconstruct the average of the effective updates. This introduces a coupling term that biases the global model:

$$\bar{B}\,\bar{A} = \frac{1}{K}\sum_{k=1}^{K}\Big[B_k A_k + (B_k - \bar{B})(A_k - \bar{A})\Big] \quad (1)$$

Here, the term $(B_k - \bar{B})(A_k - \bar{A})$ represents the interference caused by conflicting local updates. In non-IID scenarios, this deviation becomes significant, leading to biased global weights. To mitigate this, recent Federated LoRA approaches aim to isolate these conflicts by retaining specific parameters locally. Current methods implement this by freezing one factor (e.g., (Zhang et al., 2023)) or by selectively sharing specific matrices (e.g., (Guo et al., 2025; Wang et al., 2026)). Yet, these methods typically enforce a uniform splitting rule across the entire network. This rigid strategy fails to address the asymmetric heterogeneity in medical segmentation. Mathematically, the LoRA update $\Delta W_k = B_k A_k$ is driven by structurally distinct statistical dependencies across modules:

$$\Delta W_k \sim \begin{cases} \mathcal{F}_{enc}\big( B_k A_k \mid P_k(\mathbf{x}) \big) & \text{Encoder (Covariate Shift)} \\ \mathcal{G}_{dec}\big( B_k A_k \mid P_k(\mathbf{y}|\mathbf{x}) \big) & \text{Decoder (Concept Shift)} \end{cases}$$
$$(2)$$

The encoder LoRA is primarily sensitive to acquisition shifts (changes in input distribution $P(\mathbf{x})$), whereas the decoder LoRA confronts supervision variations (changes in conditional distribution $P(\mathbf{y}|\mathbf{x})$). Applying a uniform split ignores these distinct statistical roles, leaving site-specific biases entangled with shared anatomical knowledge. To this end, we aim to develop a structure-aware tuning framework that moves beyond homogeneous constraints. This raises the following challenge:

> *How to structurally disentangle the adaptation strategy to reconcile opposing heterogeneity shifts?*

Instead of enforcing a uniform splitting rule across the entire network (Zhang et al., 2023; Guo et al., 2025; Wang et al., 2026), we distinguish the roles of the encoder and decoder, modeling the former as being dominated by covariate (appearance) shift and the latter by concept (supervision) shift. Our theoretical analysis (Proposition 1, Section 3) formally derives a shift-dependent preference: minimizing the reconstruction error under covariate shift necessitates

localizing the input-side factor ($A$), whereas concept shift requires localizing the output-side factor ($B$). Guided by this theoretical derivation, we propose Inverse Asymmetric Tuning (IAT) to explicitly align the tuning strategy with these distinct structural requirements. Empirical results, illustrated in Figure 5, corroborate our theoretical analysis, confirming that this structure-aware calibration significantly outperforms uniform baselines. Guided by this analysis, IAT implements an inverse allocation strategy. Specifically, it selectively personalizes the input-projection factors ($A$) in the encoder to absorb covariate (acquisition) shifts, while personalizing the output-mapping factors ($B$) in the decoder to adapt to concept (supervision) shifts. This design explicitly aligns the parameter optimization focus with the dominant heterogeneity source of each module.

However, structural separation alone does not guarantee optimization independence. While IAT correctly allocates the parameters, the bilinear dependency of LoRA ($\Delta W = BA$) induces an intrinsic coupling during training. As gradients backpropagate through this product, the update direction of the shared matrix is inevitably modulated by the local matrix, creating a channel for implicit leakage. Consequently, the distinct subspaces defined by IAT may gradually collapse, allowing site-specific biases to contaminate the shared global model. This raises the second challenge:

> *How to ensure the functional independence of the decoupled subspaces to prevent optimization interference?*

To address this, we analyze the gradient dynamics of the bilinear interaction (Proposition 2, Section 3). Our analysis demonstrates that without constraints, the shared and local updates naturally tend toward collinearity. To counteract this, we propose the Subspace Orthogonality Regularizer (SOR). SOR imposes a soft geometric constraint that forces the shared and local subspaces to remain orthogonal. This mechanism rectifies the gradient flow, ensuring that the shared model aggregates only universal representations while site-specific variations remain strictly isolated. Our main contributions are summarized as follows:

❶ **Re-examining LoRA Allocation in FL Medical Segmentation.** Our findings indicate that the prevalent uniform splitting strategy exhibits a severe structural mismatch in medical segmentation. We reveal that the optimal parameter sharing preference is not static, but structurally inverts between the encoder and decoder due to their opposing heterogeneity patterns.

❷ **Novel Structure-Aware Framework for Dual Decoupling.** Building on the phenomenon of allocation inversion, we propose a unified framework to ensure comprehensive disentanglement. We effectively address the structural misalignment and optimization coupling inherent in federated medical LoRA training.

❸ **Theoretical Guarantees and Experimental Validation.** We provide theoretical guarantees for the shift-dependent allocation preference and the necessity of orthogonality constraints. Extensive experiments demonstrate the effectiveness and robustness of our framework compared to state-of-the-art methods.

**Conflict of Interest Disclosure.** The authors declare no financial conflicts of interest related to this work.

## 2. Related Works

**Data Heterogeneity in Federated Learning.** Federated learning (FL) enables collaborative training without sharing raw data, yet real-world deployments typically suffer from non-i.i.d. distributions (Huang et al., 2022; Hsieh et al., 2020). This statistical heterogeneity manifests primarily in three forms: label distribution skew (Kairouz et al., 2021), feature distribution skew (Huang et al., 2023), and quantity skew (Wang et al., 2020; Li et al., 2020a). Among these, we specifically focus on *feature skew* (appearance shift). In this setting, clients share a consistent label space but differ substantially in input distributions due to variations in acquisition devices or environmental conditions, leading to severe representation drift. Strategies to mitigate such heterogeneity broadly fall into two paradigms: *optimization enhancement* and *personalized adaptation*. The former stabilizes global training by rectifying local optimization dynamics, for instance, by restricting client drift via proximal regularization (Li et al., 2020a) or correcting update bias using control variates (Li et al., 2021a). However, since enforcing a single global model often proves insufficient under severe feature skew, the latter paradigm pursues personalized federated learning (PFL) (Yang et al., 2024). These approaches abandon the one-model-fits-all constraint by structurally isolating domain-specific components, such as decoupling prediction heads (Collins et al., 2021) or maintaining local normalization statistics (Li et al., 2021b), thereby allowing effective adaptation to diverse local distributions. Such feature heterogeneity is particularly pronounced in medical image segmentation, where diverse acquisition protocols create distinct visual styles that degrade model performance (Jiang et al., 2022; Zhu et al., 2024; Shao et al., 2025; Zhao et al., 2026). Recently, foundation models like Segment Anything (SAM) (Kirillov et al., 2023; Zhao et al., 2024) have emerged as powerful backbones due to their robust general-purpose representations (Liu et al., 2024; Asokan et al., 2024). However, integrating these large models into federated pipelines presents a dual challenge: achieving parameter-efficient adaptation under strict communication constraints while robustly handling significant cross-site appearance shifts (Wang et al., 2026; Hu et al., 2025b).

**LoRA in Federated Learning.** Foundation models have recently become a strong backbone choice in federated learning (Fan et al., 2023), but full-parameter fine-tuning is often impractical due to the high communication and on-device training cost (Xu et al., 2024). Parameter-efficient fine-tuning (PEFT) addresses this by freezing the backbone and updating only a small parameter subset (Houlsby et al., 2019; Liu et al., 2022; Hu et al., 2024). Notably, Low-Rank Adaptation (LoRA) (Hu et al., 2022) is widely favored in FL for its communication efficiency, achieved by transmitting only lightweight low-rank factors (Yi et al., 2023; Liu et al., 2025; Long et al., 2024). However, the bilinear nature of LoRA ($\Delta W = BA$) complicates aggregation: simply averaging local factors at the server fails to reconstruct the average effective update, a discrepancy that worsens under non-i.i.d. data (Sun et al., 2024). To mitigate this, distinct aggregation methods have been proposed. One stream employs *asymmetric sharing* (Guo et al., 2025; Zhang et al., 2023) or *residual reparameterization* (Yan et al., 2025), selectively aggregating factors or restructuring updates to minimize cross-client interference. Another stream focuses on *adaptive configurations*, such as importance-aware splitting rules (Liao et al., 2025; Zhao et al., 2025) or heterogeneous rank allocations (Cho et al., 2024; Chen et al., 2024; Wang et al., 2024), to handle diverse client constraints. It is worth noting that most current Federated LoRA strategies are developed for decoder-only Large Language Models (LLMs). Conversely, medical segmentation employs encoder–decoder architectures (e.g., SAM) with dense pixel-level supervision, creating distinct structural constraints. Consequently, directly applying LLM-centric protocols may be suboptimal, particularly as heterogeneity tends to affect the encoder and decoder modules in different ways.

## 3. Methodology

### 3.1. Preliminary

**Federated Medical Segmentation.** We consider a federated segmentation task with $K$ clients. Each client $k$ holds a private dataset $\mathcal{D}_k = \{(x_i, y_i)\}_{i=1}^{N_k}$ drawn from a distinct distribution $\mathcal{P}_k$. The goal is to minimize the global objective $\min_\Theta \sum_{k=1}^{K} p_k \mathcal{L}_k(\Theta)$. We formulate the segmentation network $\mathcal{F}$ as a composition of an Encoder ($\mathcal{E}$) and a Decoder ($\mathcal{D}$), i.e., $\mathcal{F} = \mathcal{D} \circ \mathcal{E}$. This architecture faces dual heterogeneity in medical imaging: (i) Acquisition Shift ($P_k(x)$), caused by varying scanner protocols, primarily affects the Encoder which extracts low-level features; (ii) Supervision Shift ($P_k(y|x)$), caused by distinct annotation standards, primarily affects the Decoder which maps features to semantic predictions.

**Low-Rank Adaptation (LoRA).** To efficiently adapt the pre-trained parameters $W_0 \in \mathbb{R}^{d_{out} \times d_{in}}$ (in both $\mathcal{E}$ and $\mathcal{D}$), LoRA introduces a low-rank update $\Delta W = BA$, where

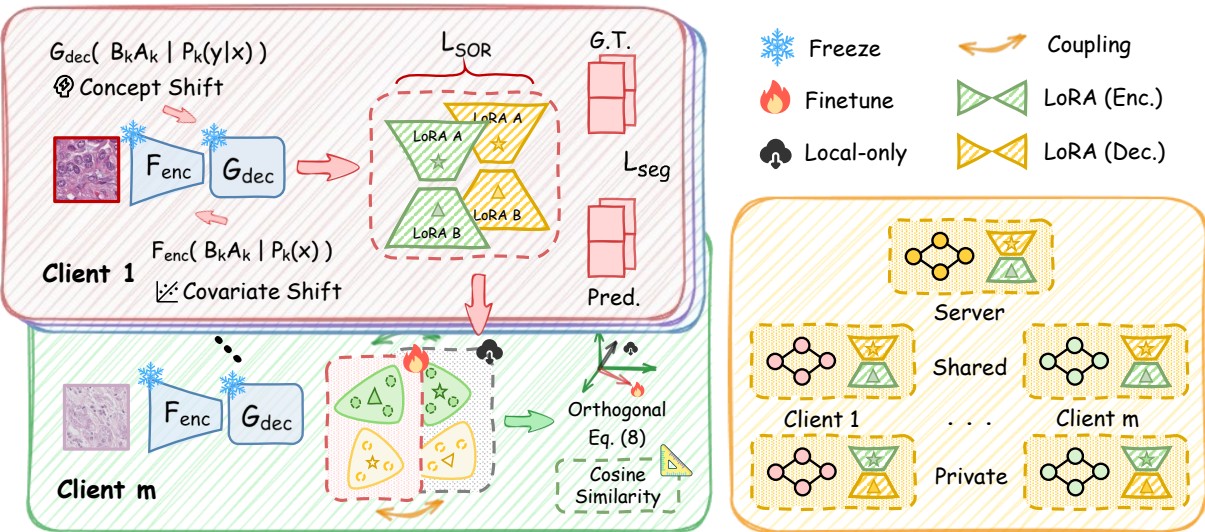

*Figure 2.* **Architecture illustration** of the proposed framework. To address the structural misalignment between covariate and concept shifts, we implement **Inverse Asymmetric Tuning (IAT)** which allocates trainable LoRA factors inversely across the encoder and decoder. To prevent the coupling of shared knowledge with local biases, we apply a **Subspace Orthogonality Regularizer (SOR)** during local updates. The server aggregates only the shared components to derive a generalized global model while preserving local personalization. Zoom in for details.

$B \in \mathbb{R}^{d_{out} \times r}$ and $A \in \mathbb{R}^{r \times d_{in}}$ ($r \ll d_{in}, d_{out}$). The forward pass is:

$$h = W_0 x + BAx, \tag{3}$$

where $x$ denotes the input to the layer. Here, $A$ acts as the input projector compressing information into the rank-$r$ subspace, while $B$ acts as the output projector reconstructing it back to the functional dimension.

### 3.2. Proposed Method

To address the critical issues of **structural mismatch** in parameter allocation and **implicit optimization coupling** under dual heterogeneity, we propose a dual-decoupling framework for federated medical segmentation. As illustrated in Figure 2, FedIAT effectively disentangles general and site-specific knowledge by integrating **Inverse Asymmetric Tuning (IAT)** to structurally align LoRA factors with module-specific shifts, and a **Subspace Orthogonality Regularizer (SOR)** to functionally suppress gradient leakage during training. This synergy ensures that global knowledge and site-specific priors are decoupled in both structure and optimization dynamics.

#### 3.2.1. INVERSE ASYMMETRIC TUNING

**Motivation.** *The optimal parameter allocation strategy in federated segmentation is not uniform but structurally inverts depending on the dominant heterogeneity type.* Existing PFL-LoRA methods typically enforce a uniform splitting rule, ignoring the structural asymmetry in medical seg-

mentation. The encoder is primarily exposed to acquisition shifts (input domain), whereas the decoder faces supervision shifts (output domain). To rigorously explain why this necessitates a flexible strategy, we analyze the approximation error of a linear surrogate layer.

**Proposition 3.1** (Shift-Dependent Preference)**.** *Consider a linear surrogate layer $y = (W_0 + BA)x$ with rank $r$. (i) Under Covariate Shift (input subspace rotation $x_k = R_k x_{gen}$), minimizing the approximation error favors sharing $B$ and personalizing $A_k$ to align the client-specific input row space. (ii) Under Concept Shift (target mapping rotation $y_k = T_k y_{gen}$), minimizing the error favors sharing $A$ and personalizing $B_k$ to align the client-specific output column space.*

*Proof.* See Appendix A.1. □

**Method.** Guided by Proposition 1 and the empirical "crossover" pattern, we propose **Inverse Asymmetric Tuning (IAT)**, which discards uniform splitting in favor of a module-aware protocol. Specifically, let $\mathcal{E}$ and $\mathcal{D}$ denote the sets of LoRA-adapted layers in the encoder and decoder, respectively.

For encoder layers ($l \in \mathcal{E}$), dominated by acquisition shift, we adopt a Local-$A$ / Shared-$B$ strategy. Clients optimize the input projector $A_k$ locally to filter site-specific imaging artifacts, while the output projector $B$ is aggregated. The server update rule is:

$$B_{agg}^{(t+1,l)} \leftarrow \sum_{k=1}^{K} p_k B_k^{(t+1,l)}, \quad \forall l \in \mathcal{E}, \tag{4}$$

while the input projector $A_k^{(t+1,l)}$ is retained locally.

Conversely, for decoder layers ($l \in \mathcal{D}$), dominated by supervision shift, we invert this strategy to Shared-$A$ / Local-$B$. Here, the input projector $A$ is aggregated to maintain a consistent shared feature subspace:

$$A_{agg}^{(t+1,l)} \leftarrow \sum_{k=1}^{K} p_k A_k^{(t+1,l)}, \quad \forall l \in \mathcal{D}, \tag{5}$$

while the output projector $B_k^{(t+1,l)}$ is retained locally to adapt to divergent annotation standards. This structural inversion explicitly aligns parameter roles with the source of heterogeneity.

### 3.2.2. SUBSPACE ORTHOGONALITY REGULARIZER

**Motivation.** *Structural decoupling alone is insufficient; the bilinear parameterization induces implicit gradient coupling, necessitating functional constraints to prevent leakage.* Although IAT structurally separates parameters, the optimization dynamics remain entangled due to the multiplicative nature of LoRA ($\Delta W = BA$). The gradient update for the shared factor is functionally dependent on the local factor ($G_k A_k^\top$), creating a channel for heterogeneity leakage. We formalize this interaction through standard SGD decomposition:

**Proposition 3.2** (Bilinear Leakage in Federated Updates). *Let $\Delta W_k = BA_k$ where $B$ is shared and $A_k$ is local. The aggregated global update for $B$ with learning rate $\eta$ can be decomposed as:*

$$B^{(t+1)} = B^{(t)} - \eta \underbrace{\sum_{k=1}^{K} p_k G_k \bar{A}^\top}_{\text{Common Drift}} - \eta \underbrace{\sum_{k=1}^{K} p_k G_k (A_k - \bar{A})^\top}_{\text{Heterogeneity Leakage}}, \tag{6}$$

*where $\bar{A} = \sum p_k A_k$, and $G_k := \nabla_{\Delta W} \mathcal{L}_k$ denotes the gradient w.r.t. the adaptation matrix $\Delta W$. The last term represents the leakage where local deviations ($A_k - \bar{A}$) contaminate the shared update. An analogous decomposition holds symmetrically for the decoder (shared $A$, local $B$) by swapping the roles of $(A, B)$.*

*Proof.* See Appendix A.2. □

**Method.** To suppress this leakage, we introduce SOR, which discourages alignment between the shared update direction and the local drift direction in a compact $r \times r$ proxy space. Here, let $A_{0,k}^{(t,l)}, B_{0,k}^{(t,l)}$ denote the detached parameter anchors at the start of round $t$, and let $\delta A_k^{(t,l)}, \delta B_k^{(t,l)}$ be the exponential moving averages (EMA) of the private-factor drifts within the round. We define the rank-efficient proxies

with explicit stop-gradient (sg) operations as:

$$\begin{aligned}
P_{sh,k}^{(t,l)} &= \left(B_k^{(t,l)} - B_{0,k}^{(t,l)}\right)^\top B_{0,k}^{(t,l)}, \\
P_{lo,k}^{(t,l)} &= \text{sg}\left[A_{0,k}^{(t,l)}\left(\delta A_k^{(t,l)}\right)^\top\right], \quad \forall l \in \mathcal{E}; \\
Q_{sh,k}^{(t,l)} &= \left(A_k^{(t,l)} - A_{0,k}^{(t,l)}\right) A_{0,k}^{(t,l)\top}, \\
Q_{lo,k}^{(t,l)} &= \text{sg}\left[B_{0,k}^{(t,l)\top} \delta B_k^{(t,l)}\right], \quad \forall l \in \mathcal{D}.
\end{aligned} \tag{7}$$

We then minimize the squared normalized Frobenius inner product between these proxies. By minimizing Eq. (8), SOR produces gradients primarily for the shared factors, ensuring they evolve orthogonally to the local drifts while leaving personalization unconstrained:

$$\begin{aligned}
\mathcal{L}_{\text{SOR}}^{(k)} &= \sum_{l \in \mathcal{E}} \left( \frac{\langle P_{sh,k}^{(t,l)}, P_{lo,k}^{(t,l)} \rangle_F}{\|P_{sh,k}^{(t,l)}\|_F \|P_{lo,k}^{(t,l)}\|_F + \epsilon} \right)^2 \\
&+ \sum_{l \in \mathcal{D}} \left( \frac{\langle Q_{sh,k}^{(t,l)}, Q_{lo,k}^{(t,l)} \rangle_F}{\|Q_{sh,k}^{(t,l)}\|_F \|Q_{lo,k}^{(t,l)}\|_F + \epsilon} \right)^2.
\end{aligned} \tag{8}$$

Each client $k$ optimizes the total objective $\mathcal{L}_{\text{total}}^{(k)} = \mathcal{L}_{\text{seg}} + \lambda \mathcal{L}_{\text{SOR}}^{(k)}$, and the server aggregates factors according to the IAT protocol.

### 3.3. Convergence Analysis

We establish the convergence guarantees of the proposed framework within the standard non-convex federated optimization analysis. To explicitly model the partial sharing mechanism, we parameterize the full optimization space as $\Theta := (\Theta^{\text{sh}}, \{\Theta_k^{\text{lo}}\}_{k=1}^K)$, where $\Theta^{\text{sh}}$ represents the aggregated LoRA factors (shared across clients) and $\Theta_k^{\text{lo}}$ denotes the client-specific local factors. The global objective is formulated as:

$$\min_\Theta F(\Theta) := \sum_{k=1}^{K} p_k \mathcal{L}_k(\Theta_k) \tag{9}$$

here the local objective is defined as $\mathcal{L}_k(\Theta_k) := \mathcal{L}_k^{\text{seg}}(\Theta^{\text{sh}}, \Theta_k^{\text{lo}}) + \lambda \mathcal{L}_k^{\text{SOR}}(\Theta_k)$. Here, $p_k$ is the relative weight of client $k$. The training process spans $R$ communication rounds. In each round, participating clients perform $E$ steps of local SGD with stepsize $\eta$ to update both shared and local components, after which the server aggregates only $\Theta^{\text{sh}}$.

We adopt the following standard assumptions to characterize the optimization landscape.

**Assumption 3.1** ($L$-Smoothness). For each client $k$, the local objective $\mathcal{L}_k$ is differentiable and $L$-smooth with respect to $\Theta_k$. That is, for any parameters $\Theta_1, \Theta_2$:

$$\mathcal{L}_k(\Theta_1) \leq \mathcal{L}_k(\Theta_2) + \langle \nabla \mathcal{L}_k(\Theta_2), \Theta_1 - \Theta_2 \rangle + \frac{L}{2}\|\Theta_1 - \Theta_2\|_F^2. \tag{10}$$

**Assumption 3.2** (Bounded Gradients). Let $g_{k,t} = \nabla \mathcal{L}_k(\Theta_{k,t}; \xi_{k,t})$ be the unbiased stochastic gradient sampled from client $k$ at step $t$. The expected squared norm is uniformly bounded, i.e., $\mathbb{E}\|g_{k,t}\|_F^2 \leq G^2$ for all $k, t$.

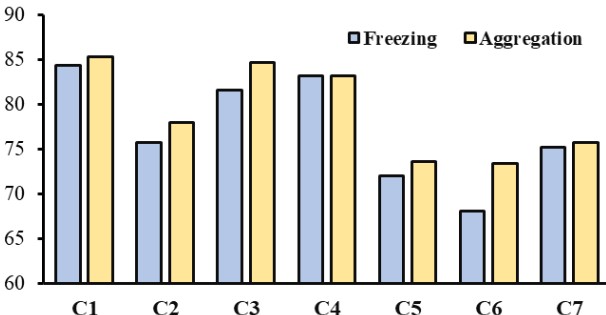

*Figure 3.* **Preliminary ablation on LoRA configuration.** We compare the performance of applying LoRA to the *Encoder-only* versus the *Encoder+Decoder*. The results demonstrate that, unlike classification tasks, medical segmentation requires adapting the decoder to reconstruct precise pixel-level details, validating our choice to inject LoRA into both modules.

**Assumption 3.3** (Non-Degenerate LoRA Factors)**.** To ensure sufficient gradient flow through the low-rank bottleneck, we assume the LoRA factors remain non-degenerate (i.e., $\sigma_{\min} \geq \delta > 0$). Consequently, there exist constants $c_A, c_B > 0$ such that the projection of the true gradient onto the local update direction $U_{k,t}$ satisfies:

$$\langle \nabla \mathcal{L}_k(\Theta_{k,t}), U_{k,t} \rangle \geq (c_A + c_B)\|\nabla \mathcal{L}_k(\Theta_{k,t})\|_F^2, \quad (11)$$

where the inner product is defined over the corresponding LoRA-parameter subspace.

Assumptions 3.1 and 3.2 are widely adopted in federated optimization (Li et al., 2020b). Assumption 3.3 is specific to low-rank parameterization, guaranteeing that the update direction remains a valid descent direction by preventing the collapse of the optimization landscape.

Based on these assumptions, we derive the following convergence rate (detailed proof in Appendix A.3).

**Theorem 3.1.** *Let Assumptions 3.1–3.3 hold. Let $T$ be the total number of local iterations. By setting the learning rate $\eta = \min\{\bar{\eta}, \sqrt{\frac{4D}{MT}}\}$, where $D$ bounds the initial suboptimality and $M = 2LC_2G^2$ is a problem-dependent constant (defined in Appendix), we have:*

$$\frac{1}{KT} \sum_{k=1}^{K} \sum_{t=0}^{T-1} \mathbb{E}\left[\|\nabla \mathcal{L}_k(\Theta_{k,t})\|_F^2\right] \leq \frac{2}{c_A + c_B} \sqrt{\frac{DM}{T}}. \quad (12)$$

Theorem 3.1 indicates that our method achieves an $\mathcal{O}(1/\sqrt{T})$ convergence rate to a stationary point. This matches the standard rate of FedAvg in non-convex settings up to lower-order aggregation drift terms (see Appendix), confirming that our asymmetric partial sharing strategy preserves theoretical convergence guarantees.

## 4. Experiments

### 4.1. Experiment Settings

**Datasets**. To comprehensively evaluate the effectiveness of our proposed framework, we conduct extensive experiments on challenging medical image segmentation benchmarks under federated learning settings. We utilize two widely used and publicly available tasks, as detailed below.

- **Histology nuclei segmentation:** We evaluate binary nuclei segmentation in a federated, cross-domain setting comprising seven clients. The setup includes four clients derived from specific tissue types within the Pan-Nuke dataset (Adrenal gland, Esophagus, Bile duct, and Uterus) (Gamper et al., 2019; 2020). The remaining three clients correspond to the independent MoNuSeg (Kumar et al., 2017), MoNuSAC2020 (Verma et al., 2021), and TNBC (Naylor et al., 2018) datasets. This benchmark is characterized by significant heterogeneity across clients in terms of tissue origin, staining procedures, and scanner variations between different centers. To ensure a unified binary task, all instance masks were converted to foreground/background masks.

- **Fundus photography images segmentation**: This task involves the joint segmentation of the optic disc (OD) and optic cup (OC) across four decentralized clients representing different public fundus datasets: REFUGE (Orlando et al., 2020), ORIGA-light (Zhang et al., 2010), Drishti-GS1 (Sivaswamy et al., 2015), and G1020 (Bajwa et al., 2020). Client heterogeneity primarily arises from diverse image acquisition conditions across different clinical sites, variations in patient cohorts, and distinct imaging protocols (e.g., the specific 30° field-of-view in Drishti-GS1 versus standard 45° protocols).

We conduct experiments in a federated setting by treating each dataset as an individual client, mimicking real-world scenarios where each medical institution acts as a distinct participant. While these clients share a unified label space, their image appearances vary significantly due to diverse image acquisition protocols and equipment specifications. Following (Liu et al., 2024; Wang et al., 2026), we adopt the same preprocessing pipelines and data splitting protocols.

**Implementation Details.** We present our implementation details from three aspects:

- **Model:** We adopt the Segment Anything Model (SAM) (Kirillov et al., 2023) as our foundation. Specifically, we utilize the **SAM ViT-B** variant initialized with weights pre-trained on the SA-1B dataset. The input images are resized to $1024 \times 1024$ and normalized following standard SAM protocols. Our framework is implemented in PyTorch and executed on NVIDIA A100 GPUs. For the federated setting, the training procedure spans 200 global communication rounds. In each round, the server employs

*Table 1.* **Comparison with State-of-the-Art Fine-Tuning Solutions** on Histology nuclei and Fundus photography images. The optimal and sub-optimal results are denoted by boldface and underlining. ↑ means improved accuracy compared with the sub-optimal results.

| Methods | Histology nuclei | | | | | | | | Fundus photography images | | | | |
|---|---|---|---|---|---|---|---|---|---|---|---|---|---|
| | ADRENA | ESOPHAGUS | BILE-DUCT | UTERUS | MONUSAC | TNBC | MONUSEG | *Avg* | REFUGE | ORIGA | G1020 | DRISHTI-GS1 | *Avg* |
| *LoRA Rank=8* | | | | | | | | | | | | | |
| FedIT | 85.25 | **78.98** | 84.69 | 83.28 | 72.97 | 72.16 | 76.42 | 79.11 | 88.85 | 84.60 | 78.15 | 57.74 | 77.33 |
| FLoRA | 79.45 | 71.67 | 77.09 | 79.77 | 60.49 | 63.84 | 65.24 | 71.08 | 79.89 | 80.98 | 69.50 | 71.71 | 75.52 |
| FedSA | 84.89 | 77.07 | 83.97 | 82.29 | 75.68 | 75.65 | 81.10 | 80.09 | 88.69 | 83.67 | **79.15** | 80.64 | 83.04 |
| FFA-LoRA | 82.21 | 69.99 | 77.76 | 80.30 | 65.40 | 3.59 | 0.74 | 54.29 | 89.27 | 81.12 | 72.60 | 29.07 | 68.02 |
| FedDPA | 83.55 | 76.10 | 81.65 | 81.35 | 74.36 | 76.93 | **81.51** | 79.35 | 88.08 | 84.70 | 72.75 | 78.28 | 80.95 |
| LoRA-FAIR | 84.13 | 75.64 | 82.73 | 83.10 | 72.50 | 65.70 | 75.00 | 76.97 | 89.24 | 84.78 | 77.86 | 70.78 | 80.67 |
| FlexLoRA | 85.71 | 77.30 | 82.88 | **84.13** | 74.35 | 70.73 | 72.36 | 78.21 | **89.38** | 85.40 | 77.53 | 65.72 | 79.51 |
| FRLoRA | 83.87 | 76.74 | 81.39 | 83.16 | 72.74 | 70.48 | 71.35 | 77.11 | 88.90 | 84.69 | 77.21 | 60.55 | 77.83 |
| **Ours** | **85.80** | 77.10 | **85.55** | 82.79 | **78.01** | **79.49** | 81.06 | **81.40** | 89.00 | **85.47** | 78.15 | **85.43** | **84.52** |
| *LoRA Rank=16* | | | | | | | | | | | | | |
| FedIT | 84.48 | 76.81 | 85.03 | 83.25 | 74.93 | 77.79 | 77.49 | 79.97 | **89.67** | 85.80 | 80.00 | 41.52 | 74.25 |
| FLoRA | 83.06 | 72.76 | 79.03 | 81.30 | 63.68 | 59.62 | 68.29 | 72.53 | 83.44 | 83.37 | 71.99 | 74.84 | 78.41 |
| FedSA | 84.13 | 78.20 | 83.39 | 81.66 | 75.62 | 78.42 | **82.86** | 80.61 | 89.39 | 86.04 | 79.36 | 83.91 | 84.68 |
| FFA-LoRA | 81.96 | 69.41 | 77.23 | 80.00 | 63.80 | 3.03 | 0.87 | 53.76 | 89.34 | 80.95 | 72.46 | 26.82 | 67.39 |
| FedDPA | 84.17 | 77.64 | 83.53 | 82.47 | 75.63 | 76.95 | 81.75 | 80.31 | 88.76 | 84.61 | 73.65 | 77.42 | 81.11 |
| LoRA-FAIR | 84.23 | 78.13 | 84.32 | 83.41 | 75.38 | 74.25 | 77.70 | 79.63 | 88.73 | **86.51** | 75.46 | 64.78 | 78.87 |
| FlexLoRA | **85.77** | **79.52** | 81.90 | 83.32 | 74.83 | 71.11 | 74.01 | 78.64 | 89.48 | 86.19 | 78.35 | 71.87 | 81.47 |
| FRLoRA | 84.43 | 78.01 | 80.94 | **83.70** | 74.90 | 71.11 | 74.20 | 78.18 | 89.42 | 84.75 | 78.78 | 59.77 | 78.18 |
| **Ours** | 85.69 | 78.19 | **85.52** | 83.51 | **75.96** | **80.49** | 81.12 | **81.49** | 89.61 | 86.48 | **80.30** | **86.43** | **85.70** |

*Table 2.* **Ablation study of key components** on the retinal fundus segmentation task.

| IAT | SOR | Fundus photography images | | | | |
|---|---|---|---|---|---|---|
| | | REFUGE | ORIGA | G1020 | DRISHTI-GS1 | *Avg* |
| LoRA | | 88.85 | 84.60 | 78.15 | 57.74 | 77.33 |
| ✓ | | **89.13** | 85.39 | 77.71 | 81.64 | 83.47 |
| ✓ | ✓ | 89.00 | **85.47** | **78.15** | **85.43** | **84.52** |

*Table 3.* **System Efficiency Comparison** reporting Trainable Parameters and Per-round Communicated Parameters.

| Method | Trainable Param. | Per-round Communicated Param. |
|---|---|---|
| FedIT | 0.39M | 0.78M |
| FLoRA | 4.35M | 29.94M |
| FedSA | 0.39M | 0.25M |
| FFA-LoRA | 0.26M | 0.53M |
| FedDPA | 4.65M | 8.71M |
| LoRA-FAIR | 0.39M | 0.78M |
| FlexLoRA | 0.39M | 0.78M |
| FRLoRA | 0.29M | 21.82M |
| **Ours** | 0.39M | 0.55M |

a **weighted aggregation strategy** (based on client sample sizes) to update the global model. Locally, each client performs training for 1 epoch using the Adam optimizer with a batch size of 4. The optimizer's momentum parameters are set to $\beta_1 = 0.9$ and $\beta_2 = 0.999$. All clients share identical hyperparameter configurations to ensure fair comparisons.

- **LoRA Configuration:** We inject LoRA rank decomposition matrices into the query and value projection layers of the Transformer blocks. Crucially, we apply LoRA to both the image encoder and the mask decoder. Unlike other tasks where the encoder dominates, medical image segmentation is a dense prediction task that relies heavily on the decoder to reconstruct pixel-level details

from high-level features. To validate this, we conducted a preliminary ablation (see Fig. 3), comparing freezing the backbone while fine-tuning LoRA on the encoder-only versus the encoder-decoder. The results demonstrate that adapting the decoder is essential for capturing boundary details, yielding superior performance.

- **Evaluation Metric:** Following prior works (Jiang et al., 2022), we employ the **Dice Similarity Coefficient (DSC)** to quantitatively evaluate the segmentation accuracy. The DSC measures the overlap between the predicted segmentation mask and the ground truth.

**Compared Baselines:** We compare our proposed framework with several state-of-the-art Federated Parameter-Efficient Fine-Tuning (FedPEFT) methods: (1) **FedIT** (Zhang et al., 2024). (2) **FLoRA** (Wang et al., 2024). (3) **FedSA** (Guo et al., 2025). (4) **FFA-LoRA** (Sun et al., 2024). (5) **FedDPA** (Long et al., 2024). (6) **LoRA-FAIR** (Bian et al., 2025). (7) **FlexLoRA** (Bai et al., 2024). (8) **FRLoRA** (Yan et al., 2025). Note that since these methods were originally designed for Large Language Models (LLMs), we adapted their core mechanisms to our segmentation backbone to ensure a fair comparison.

### 4.2. Comparison to State-of-the-Arts

Table 1 presents the quantitative performance comparison on the Histology nuclei and Fundus photography datasets. Our results demonstrate that the proposed framework achieves superior performance across diverse heterogeneous clients. Existing FedPEFT methods often struggle to effectively align divergences caused by severe domain shifts, leading to performance instability. For instance, FFA-LoRA suffers from catastrophic degradation on specific clients (e.g.,

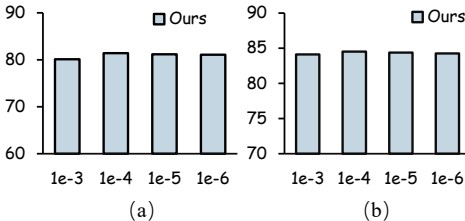

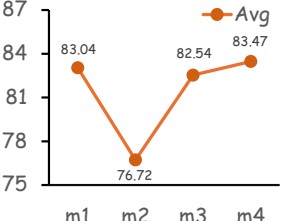

*Figure 4.* **Sensitivity analysis of** $\lambda$**.** Left: Histology nuclei. Right: Fundus photography.

*Figure 5.* **Empirical validation of theoretical analysis.** We compare different splitting configurations labeled as M1 to M4. **M1**: Uniform Strategy (e.g., FedSA). **M2**: Inappropriate Uniform Strategy. **M3**: Reverse Hybrid Strategy. **M4**: Proposed Inverse Asymmetric Tuning (IAT). The results show that M4 consistently outperforms other heuristic combinations, validating the shift-dependent structural preference.

3.59% on TNBC), while FedIT fails to generalize to distinct domains like Drishti-GS1 (57.74%). In contrast, our method successfully maintains robust generalization capabilities under these challenging conditions. By effectively decoupling covariate and concept shifts, our framework achieves the highest average DSC of **81.40%** and **84.52%** on the two tasks, surpassing the second-best methods by **1.31%** and **1.48%**, respectively. Specifically, on the most distinct domain (Drishti-GS1), our method outperforms the baseline FedIT by a remarkable margin of over 27%, validating the effectiveness of our structure-aware tuning strategy in handling medical heterogeneity.

### 4.3. Diagnostic Analysis

**Hyper-parameter Study**. Fig. 4 illustrates the performance variations across both **Histology and Fundus datasets**. We observe a consistent trend on both modalities: introducing the regularization term ($\lambda > 0$) improves performance compared to negligible weighting ($10^{-6}$), confirming the necessity of the proposed SOR module. Specifically, the performance peaks at $\lambda = 10^{-4}$, achieving optimal DSC scores of 81.40% (Histology) and 84.52% (Fundus). While an overly large $\lambda$ (e.g., $10^{-3}$) leads to a slight decline due to over-regularization, the model maintains robust performance within the optimal range. Therefore, we universally set $\lambda = 10^{-4}$ for all tasks to ensure stable convergence.

**Ablation of Key Component**. We evaluate the contribution of each component using four diverse fundus datasets. Table 2 presents the results, demonstrating that both components are essential. The vanilla Federated LoRA struggles with severe domain shifts (e.g., 57.74% on DRISHTI-GS1). Incorporating **IAT** significantly mitigates this, boosting the average DSC to 83.47%. Furthermore, adding **SOR** promotes the orthogonality between shared and local features, yielding the optimal average performance of 84.52%.

**Empirical Validation of Theoretical Analysis.** We conducted a pilot study on the Fundus dataset to validate the shift-dependent preference derived in Proposition 3.1. We compared our proposed structure-aware strategy with standard *Uniform Strategies* (e.g., FedSA) and other heuristic combinations. As shown in Figure 5, our method consistently outperforms other configurations, achieving an optimal DSC of 83.47%. Specifically, compared to the strong

baseline FedSA (83.04%), our approach yields a clear improvement by explicitly assigning local modules to absorb specific shifts. Moreover, deviating from this theoretical guidance—such as reversing the allocation rule—leads to suboptimal performance (82.54% or lower). These empirical results strongly support our theorem: maximizing parameter efficiency requires inverting the tuning strategy between the encoder and decoder to align with their dominant heterogeneity types.

**System Efficiency.** Table 3 details the comprehensive efficiency metrics. *(1) Parameter Efficiency:* Our method demonstrates exceptional storage efficiency. With only **0.39M** trainable parameters, it maintains a lightweight footprint comparable to FedIT and FedSA. This is in sharp contrast to FLoRA (4.35M) and FedDPA (4.65M), which require updating significantly more parameters, imposing heavy storage burdens on local devices. *(2) Communication Cost:* The proposed IAT protocol inherently optimizes bandwidth usage. By decoupling the model and transmitting only the shared LoRA factors (while keeping personalized factors local), we reduce the per-round communication cost to **0.55M**. This represents a substantial saving compared to the standard FedIT (0.78M), which aggregates all LoRA parameters. *(3) Performance-Efficiency Trade-off:* Although FedSA and FFA-LoRA achieve lower communication costs (0.25M/0.53M) through aggressive compression strategies, they fail to maintain segmentation accuracy in complex medical scenarios. In contrast, our method achieves SOTA performance with only a marginal communication increase, demonstrating the best trade-off between communication overhead and model generalization.

## 5. Conclusion

In this paper, we introduce a novel structure-aware federated tuning framework designed to adapt the Segment Anything Model (SAM) for heterogeneous medical image segmentation. Different from uniform splitting strategies, our method employs **Inverse Asymmetric Tuning** to decouple the opti-

mization landscape: it selectively personalizes the encoder to absorb covariate (appearance) shifts while adapting the decoder to align with concept (supervision) shifts. To further mitigate the interference between global and local representations, we incorporate a **Subspace Orthogonality Regularizer**, which forces the shared and personalized subspaces to remain functionally orthogonal. Extensive experiments on multi-site histology nuclei segmentation and fundus photography images segmentation datasets demonstrate that our approach achieves state-of-the-art performance with superior parameter and communication efficiency.

## Acknowledgements

This work is partially supported by the Science and Technology Innovation Program of Xiongan New Area (Grant No.2025XAGG0045), the National High Level Hospital Clinical Research Funding (2025-PUMCH-H-006), and the NTU AI-for-X Postdoctoral Fellowship.

## Impact Statement

This paper presents work whose goal is to advance the field of Machine Learning. There are many potential societal consequences of our work, none which we feel must be specifically highlighted here.

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

# A. Appendix

## A.1. Proof of Proposition 1 (Shift-Dependent Preference)

**Proposition 1 Restatement.** *Consider a linear layer target $W_k^*$ under domain shift. (i) Under Covariate Shift (input subspace rotation $x_k = R_k x_{gen}$), minimizing the rank-$r$ approximation error $\| \cdot \|_F$ favors sharing $B$ and personalizing $A_k$ (Local-A). (ii) Under Concept Shift (output subspace rotation $y_k = T_k y_{gen}$), minimizing the error favors sharing $A$ and personalizing $B_k$ (Local-B).*

*Proof.* We analyze an oracle approximation objective to isolate the structural preference of factor sharing strategies. Let $W^* \in \mathbb{R}^{d_{out} \times d_{in}}$ be the ideal generalized weight matrix with Singular Value Decomposition (SVD) $W^* = U\Sigma V^\top$. According to the Eckart-Young-Mirsky theorem, the optimal rank-$r$ approximation is uniquely determined by the principal subspaces.

**Case (i): Covariate Shift (Input Rotation).** We model subspace covariate shift as an orthogonal rotation of the input feature space. Let $x_k = R_k x$, where $R_k \in \mathbb{R}^{d_{in} \times d_{in}}$ is orthogonal. The effective client-specific weight is $W_k^* = W^* R_k^\top$. The SVD of the shifted weight is:
$$W_k^* = U\Sigma V^\top R_k^\top = U\Sigma (R_k V)^\top.$$

Observe that the Column Space $\mathrm{Col}(W_k^*) = \mathrm{span}(U)$ remains invariant, while the Row Space $\mathrm{Row}(W_k^*) = \mathrm{span}(R_k V)$ rotates with $R_k$.

- **Optimality of Shared-$B$ / Local-$A_k$:** Let the shared $B$ capture the invariant column space: $B = U_r \Sigma_r^{1/2}$ (top-$r$ left singular vectors). Client $k$ can optimally solve for $A_k$ to align with the rotated row space: $A_k = \Sigma_r^{1/2} (R_k V_r)^\top$. This yields $BA_k = U_r \Sigma_r (R_k V_r)^\top$, which exactly matches the optimal rank-$r$ approximation of $W_k^*$ given by the Eckart-Young-Mirsky theorem. The error is minimal (bounded only by the truncated singular values of $W^*$) and independent of the shift $R_k$.

- **Suboptimality of Shared-$A$ (Impossibility Result):** Assume a fixed shared $A$ of rank $r$ exists. For $A$ to support an optimal approximation for all clients, there must exist local $B_k$ such that $B_k A = (W_k^*)_r$. This necessitates the subspace containment condition:
$$\mathrm{Row}((W_k^*)_r) \subseteq \mathrm{Row}(A), \quad \forall k \in \{1, \dots, K\}.$$

However, $\mathrm{Row}((W_k^*)_r) = \mathrm{span}(R_k V_r)$. Under the generic condition that the rotations $R_k$ are non-degenerate such that the union of target row subspaces spans a dimension greater than $r$ (i.e., $\dim(\bigcup_k \mathrm{span}(R_k V_r)) > r$), no single rank-$r$ matrix $A$ can contain all target subspaces simultaneously. Consequently, a shared $A$ inevitably incurs strictly higher approximation error compared to the Shared-$B$ strategy.

*Conclusion:* Under Covariate Shift, sharing the column projector $B$ is structurally superior.

**Case (ii): Concept Shift (Output Rotation).** We model concept shift as an orthogonal rotation of the output semantic space. Let $y_k = T_k y$. The client-specific weight is $W_k^* = T_k W^*$. The SVD is:
$$W_k^* = T_k U\Sigma V^\top = (T_k U)\Sigma V^\top.$$

Here, the Row Space $\mathrm{Row}(W_k^*) = \mathrm{span}(V)$ remains invariant, while the Column Space $\mathrm{Col}(W_k^*) = \mathrm{span}(T_k U)$ rotates.

- **Optimality of Shared-$A$ / Local-$B_k$:** Let shared $A$ capture the invariant row space: $A = \Sigma_r^{1/2} V_r^\top$. Client $k$ sets $B_k = (T_k U_r)\Sigma_r^{1/2}$. This construction achieves the optimal rank-$r$ approximation error for any $T_k$.

- **Suboptimality of Shared-$B$ (Impossibility Result):** Similarly, a shared $B$ imposes the constraint $\mathrm{Col}((W_k^*)_r) \subseteq \mathrm{Col}(B)$. Since $\mathrm{Col}((W_k^*)_r) = \mathrm{span}(T_k U_r)$, under the condition that the union of rotated column subspaces exceeds rank $r$ (due to diverse $T_k$), a fixed $B$ cannot align with all clients simultaneously.

*Conclusion:* Under Concept Shift, sharing the row projector $A$ is structurally superior. $\qquad\square$

---

**Algorithm 1** Our proposed method.

---

**Input**: clients $\{(\mathcal{D}_k, p_k)\}_{k=1}^K$, frozen backbone $W_0$, encoder layers $\mathcal{E}$, decoder layers $\mathcal{D}$, rounds $R$, local steps $E$, stepsize $\eta$, SOR weight $\lambda$, EMA momentum $\rho$, stability $\epsilon$
**Output**: shared factors $\Theta^{\mathrm{sh}}$ and local factors $\{\Theta_k^{\mathrm{lo}}\}_{k=1}^K$

1: **Init:** server $\Theta^{\mathrm{sh}} \leftarrow \{B^{(l)}\}_{l\in\mathcal{E}} \cup \{A^{(l)}\}_{l\in\mathcal{D}}$; clients $\Theta_k^{\mathrm{lo}} \leftarrow \{A_k^{(l)}\}_{l\in\mathcal{E}} \cup \{B_k^{(l)}\}_{l\in\mathcal{D}}$.
2: **for** $r = 0$ **to** $R-1$ **do**
3:     Server broadcasts $\Theta^{\mathrm{sh}}$ to participating clients $\mathcal{S}_r$.
4:     **for each** $k \in \mathcal{S}_r$ **in parallel do**
5:         Form $\Theta_k \leftarrow (\Theta^{\mathrm{sh}}, \Theta_k^{\mathrm{lo}})$ and set anchors $(A_{0,k}, B_{0,k}) \leftarrow \mathrm{sg}(A_k, B_k)$; initialize EMA drifts $(\delta A_k, \delta B_k) \leftarrow 0$.
6:         **for** $s = 1$ **to** $E$ **do**
7:             Sample mini-batch $\xi_{k,s} \sim \mathcal{D}_k$ and compute $\mathcal{L}_k^{\mathrm{seg}}(\Theta_k; \xi_{k,s})$.
8:             $\mathcal{L}_k^{\mathrm{SOR}} \leftarrow \mathrm{SOR}(\Theta_k, A_{0,k}, B_{0,k}, \delta A_k, \delta B_k; \rho, \epsilon)$.
9:             $\Theta_k \leftarrow \Theta_k - \eta \nabla_{\Theta_k} \left( \mathcal{L}_k^{\mathrm{seg}} + \lambda \mathcal{L}_k^{\mathrm{SOR}} \right)$.
10:         **end for**
11:         Upload shared parts: $\{B_k^{(l)}\}_{l\in\mathcal{E}}$ and $\{A_k^{(l)}\}_{l\in\mathcal{D}}$; keep $\Theta_k^{\mathrm{lo}}$ local.
12:     **end for**
13:     **Aggregate (IAT):** for $l \in \mathcal{E}$, $B^{(l)} \leftarrow \sum_{k\in\mathcal{S}_r} p_k B_k^{(l)}$; for $l \in \mathcal{D}$, $A^{(l)} \leftarrow \sum_{k\in\mathcal{S}_r} p_k A_k^{(l)}$.
14: **end for**
15: **return** $\Theta^{\mathrm{sh}}$ and $\{\Theta_k^{\mathrm{lo}}\}_{k=1}^K$.
16: **Subroutine** $\mathrm{SOR}(\cdot)$: **return** $\sum_{l\in\mathcal{E}} \left( \frac{\langle P_{sh}^{(l)}, P_{lo}^{(l)}\rangle_F}{\|P_{sh}^{(l)}\|_F \|P_{lo}^{(l)}\|_F + \epsilon} \right)^2 + \sum_{l\in\mathcal{D}} \left( \frac{\langle Q_{sh}^{(l)}, Q_{lo}^{(l)}\rangle_F}{\|Q_{sh}^{(l)}\|_F \|Q_{lo}^{(l)}\|_F + \epsilon} \right)^2$,
    where for $l \in \mathcal{E}$: $\delta A \leftarrow \rho\delta A + (1-\rho)(A - A_0)$, $P_{sh} = (B - B_0)^\top B_0$, $P_{lo} = \mathrm{sg}(A_0 \delta A^\top)$;
    for $l \in \mathcal{D}$: $\delta B \leftarrow \rho\delta B + (1-\rho)(B - B_0)$, $Q_{sh} = (A - A_0)A_0^\top$, $Q_{lo} = \mathrm{sg}(B_0^\top \delta B)$.

---

## A.2. Proof of Proposition 2 (Bilinear Leakage)

**Proposition 2 Restatement.** *The aggregated global update exhibits a heterogeneity-induced leakage term. For a Shared-$B$ / Local-$A$ configuration, the update is decomposed as:*

$$B^{(t+1)} = B^{(t)} - \eta \underbrace{\sum_{k=1}^K p_k G_k \bar{A}^\top}_{\text{Common Drift}} - \eta \underbrace{\sum_{k=1}^K p_k G_k (A_k - \bar{A})^\top}_{\text{Heterogeneity Leakage}}. \tag{13}$$

*Proof.* Let $\mathcal{L}_k(W)$ be the loss function for client $k$. In the LoRA parameterization $\Delta W = BA_k$, the gradient w.r.t. the shared parameter $B$ is derived via the chain rule as $\nabla_B \mathcal{L}_k = G_k A_k^\top$, where $G_k \triangleq \nabla_{\Delta W} \mathcal{L}_k$. In federated averaging (FedAvg), the server aggregates updates from $K$ clients. The update rule for $B$ is:

$$B^{(t+1)} = B^{(t)} - \eta \sum_{k=1}^K p_k (G_k A_k^\top).$$

Defining the global average $\bar{A} = \sum_{k=1}^K p_k A_k$ and decomposing $A_k^\top = \bar{A}^\top + (A_k - \bar{A})^\top$, we substitute into the update rule:

$$B^{(t+1)} = B^{(t)} - \eta \sum_{k=1}^K p_k G_k \left( \bar{A}^\top + (A_k - \bar{A})^\top \right)$$

$$= B^{(t)} - \eta \left( \sum_{k=1}^K p_k G_k \right) \bar{A}^\top - \eta \sum_{k=1}^K p_k G_k (A_k - \bar{A})^\top.$$

This explicitly separates the update into the *Common Drift* (driven by $\bar{A}$) and the *Heterogeneity Leakage*.

**Symmetric Case (Decoder: Shared-$A$ / Local-$B$).** For the decoder (Shared-$A$), the gradient is $\nabla_A \mathcal{L}_k = B_k^\top G_k$. By symmetry, let $\bar{B} = \sum p_k B_k$. The update decomposition is:

$$A^{(t+1)} = A^{(t)} - \eta \, \bar{B}^\top \underbrace{\sum_{k=1}^{K} p_k G_k}_{\text{Common Drift}} - \eta \underbrace{\sum_{k=1}^{K} p_k (B_k - \bar{B})^\top G_k}_{\text{Heterogeneity Leakage}}.$$

**Remark (Conditions for Leakage and Multi-step Extension).** 1. **Vanishing Conditions:** The leakage term vanishes if either (a) $A_k = \bar{A}$ for all $k$ (no personalization/heterogeneity), or (b) the gradients $G_k$ and local deviations $(A_k - \bar{A})$ are strictly orthogonal in expectation. In non-i.i.d. settings, neither is guaranteed, leading to persistent interference. 2. **Multi-step Local Updates:** While the derivation above assumes a single SGD step for clarity, the decomposition extends to multiple local epochs. In that case, $G_k$ represents the *accumulated pseudo-gradient* over $\tau$ local steps (i.e., $G_k \propto \Delta W_k^{(effective)}$), and the structural coupling between $G_k$ and the local adapter basis persists. $\square$

### A.3. Proof of Theorem 3.1

We follow the standard smooth non-convex SGD analysis and adapt it to the LoRA-based parametrization and our proposed update rule under the *augmented* (shared/local) formulation. Recall that each client maintains variables $\Theta_k := (\Theta^{\text{sh}}, \Theta_k^{\text{lo}})$, where $\Theta^{\text{sh}}$ denotes the shared LoRA factors aggregated by the server (encoder shares $B$, decoder shares $A$), and $\Theta_k^{\text{lo}}$ denotes the remaining client-local factors. The augmented objective is $\sum_{k=1}^{K} p_k \mathcal{L}_k(\Theta_k)$. In this appendix we bound the stationarity measure on the augmented variables by analyzing the per-step decrease of a fixed client objective, relating the module update direction to the true gradient via Assumption 3.3, and telescoping over local steps.

Fix a client $k$ and a local step $t$. For clarity we omit the client index $k$ when no confusion arises and write $\Theta_t$ for $\Theta_{k,t}$.

Within our method, a single local update on client $k$ can be written in the compact form

$$\Theta_{t+1} = \Theta_t - \eta \, U_t, \tag{14}$$

where $U_t$ is the update direction induced by the gradients with respect to the LoRA factors (both shared and local) under our proposed protocol. Note that $U_t$ is a stochastic direction depending on the mini-batch $\xi_t$.

By $L$-smoothness (Assumption 1), we have

$$\mathcal{L}_k(\Theta_{t+1}) \leq \mathcal{L}_k(\Theta_t) + \langle \nabla \mathcal{L}_k(\Theta_t), \Theta_{t+1} - \Theta_t \rangle + \frac{L}{2} \|\Theta_{t+1} - \Theta_t\|_F^2 \tag{15}$$

$$= \mathcal{L}_k(\Theta_t) - \eta \langle \nabla \mathcal{L}_k(\Theta_t), U_t \rangle + \frac{L\eta^2}{2} \|U_t\|_F^2. \tag{16}$$

Taking expectation over the mini-batch $\xi_t$ and conditioning on $\Theta_t$, we obtain

$$\mathbb{E}_t \big[ \mathcal{L}_k(\Theta_{t+1}) \big] \leq \mathcal{L}_k(\Theta_t) - \eta \, \mathbb{E}_t \big[ \langle \nabla \mathcal{L}_k(\Theta_t), U_t \rangle \big] + \frac{L\eta^2}{2} \, \mathbb{E}_t \big[ \|U_t\|_F^2 \big], \tag{17}$$

where $\mathbb{E}_t[\cdot]$ denotes the conditional expectation given $\Theta_t$. By construction of our proposed method, the update $U_t$ is a linear combination of the gradients with respect to the LoRA factors $A_{k,t}$ and $B_{k,t}$ (encoder and decoder blocks). Under Assumption 3.3, there exist constants $c_A, c_B > 0$ such that

$$\langle \nabla \mathcal{L}_k(\Theta_t), U_t \rangle \geq (c_A + c_B) \left\| \nabla \mathcal{L}_k(\Theta_t) \right\|_F^2. \tag{18}$$

Intuitively, this means that the projection of the true gradient onto the LoRA subspace used by our method is sufficiently well aligned, so the update direction is a descent direction on average.

Next, we bound the second moment of $U_t$. Since $U_t$ is composed of stochastic gradients with respect to the LoRA factors, using Assumption 2 and the boundedness of $A_{k,t}$ and $B_{k,t}$ in Assumption 3, we can find constants $C_1, C_2 > 0$ such that

$$\mathbb{E}_t \big[ \|U_t\|_F^2 \big] \leq C_1 \left\| \nabla \mathcal{L}_k(\Theta_t) \right\|_F^2 + C_2 G^2. \tag{19}$$

The precise expressions of $C_1$ and $C_2$ are not critical; they depend polynomially on $C_A, C_B$ and on the block structure of the LoRA factors in our method. Substituting (18) and (19) into (17), we obtain

$$\mathbb{E}_t\big[\mathcal{L}_k(\Theta_{t+1})\big] \leq \mathcal{L}_k(\Theta_t) - \eta(c_A + c_B)\big\|\nabla\mathcal{L}_k(\Theta_t)\big\|_F^2 + \frac{L\eta^2}{2}\Big(C_1\big\|\nabla\mathcal{L}_k(\Theta_t)\big\|_F^2 + C_2 G^2\Big) \tag{20}$$

$$= \mathcal{L}_k(\Theta_t) - \Big((c_A + c_B)\eta - \frac{LC_1\eta^2}{2}\Big)\big\|\nabla\mathcal{L}_k(\Theta_t)\big\|_F^2 + \frac{LC_2 G^2}{2}\eta^2. \tag{21}$$

We now choose the stepsize $\eta$ small enough so that

$$(c_A + c_B)\eta - \frac{LC_1}{2}\eta^2 \geq \frac{c_A + c_B}{2}\eta. \tag{22}$$

Since the left-hand side is a continuous function of $\eta$ and positive near $0$, this is guaranteed whenever

$$0 < \eta \leq \frac{c_A + c_B}{LC_1}. \tag{23}$$

Substituting (22) into the previous inequality yields

$$\mathbb{E}_t\big[\mathcal{L}_k(\Theta_{t+1})\big] \leq \mathcal{L}_k(\Theta_t) - \frac{c_A + c_B}{2}\eta\big\|\nabla\mathcal{L}_k(\Theta_t)\big\|_F^2 + \frac{LC_2 G^2}{2}\eta^2. \tag{24}$$

We now sum (24) over all local steps and all clients. Let $C_3 := LC_2 G^2$, which depends on $L, C_2$, and $G$. Let $T$ denote the total number of local updates per client and recall that $\mathcal{L}_k$ is the local objective on client $k$. Taking expectation over all stochasticity and summing over $t = 0, \ldots, T - 1$ yields

$$\sum_{t=0}^{T-1}\mathbb{E}\big[\mathcal{L}_k(\Theta_{k,t+1})\big] \leq \sum_{t=0}^{T-1}\mathbb{E}\big[\mathcal{L}_k(\Theta_{k,t})\big] - \frac{(c_A + c_B)}{2}\eta\sum_{t=0}^{T-1}\mathbb{E}\big[\|\nabla\mathcal{L}_k(\Theta_{k,t})\|_F^2\big] + \frac{1}{2}TC_3\eta^2. \tag{25}$$

The left-hand side telescopes. Note that in the Federated setting, at communication boundaries $t = nE$, the shared variables are updated via server aggregation (while local variables remain on each client), which introduces an additional *aggregation drift* term that is not captured by the per-step descent inequality above. Under standard smoothness and bounded-variance assumptions for local SGD / FedAvg, this drift can be bounded by an extra term on the order of $\mathcal{O}(\eta^2 E^2 G^2)$ (up to problem-dependent constants that quantify data heterogeneity). Since we choose $\eta \propto 1/\sqrt{T}$ and treat $E$ as a fixed constant, this contribution scales as $\mathcal{O}(1/T)$ and is thus lower-order compared to the optimization rate $\mathcal{O}(1/\sqrt{T})$. For clarity and to focus on the our method's update dynamics, we omit this lower-order drift term in the derivation below.

Thus, we obtain

$$\mathbb{E}\big[\mathcal{L}_k(\Theta_{k,T})\big] - \mathbb{E}\big[\mathcal{L}_k(\Theta_{k,0})\big] \leq -\frac{(c_A + c_B)}{2}\eta\sum_{t=0}^{T-1}\mathbb{E}\big[\|\nabla\mathcal{L}_k(\Theta_{k,t})\|_F^2\big] + \frac{1}{2}TC_3\eta^2. \tag{26}$$

Rearranging gives

$$\frac{1}{T}\sum_{t=0}^{T-1}\mathbb{E}\big[\|\nabla\mathcal{L}_k(\Theta_{k,t})\|_F^2\big] \leq \frac{2}{(c_A + c_B)\eta T}\Big(\mathbb{E}[\mathcal{L}_k(\Theta_{k,0})] - \mathbb{E}[\mathcal{L}_k(\Theta_{k,T})]\Big) + \frac{C_3}{(c_A + c_B)}\eta. \tag{27}$$

Using the fact that $\mathcal{L}_k(\Theta_{k,T}) \geq \inf_{\Theta_k}\mathcal{L}_k(\Theta_k)$ and denoting $D_k = \mathcal{L}_k(\Theta_{k,0}) - \inf_{\Theta_k}\mathcal{L}_k(\Theta_k)$, we obtain

$$\frac{1}{T}\sum_{t=0}^{T-1}\mathbb{E}\big[\|\nabla\mathcal{L}_k(\Theta_{k,t})\|_F^2\big] \leq \frac{2D_k}{(c_A + c_B)\eta T} + \frac{C_3}{(c_A + c_B)}\eta. \tag{28}$$

Finally, summing over all $K$ clients and dividing by $K$ yields

$$\frac{1}{KT}\sum_{k=1}^{K}\sum_{t=0}^{T-1}\mathbb{E}\big[\|\nabla\mathcal{L}_k(\Theta_{k,t})\|_F^2\big] \leq \frac{2D}{(c_A + c_B)\eta T} + \frac{C_3}{(c_A + c_B)}\eta, \tag{29}$$

where $D = \max_k D_k$. To obtain an explicit $\mathcal{O}(T^{-1/2})$ rate, we choose the stepsize $\eta$ to balance the two terms. Define $M := 2C_3 = 2LC_2G^2$ and recall the admissible range $0 < \eta \leq \bar{\eta}$ with $\bar{\eta} := \frac{c_A + c_B}{LC_1}$ (cf. (23)). Choosing

$$\eta = \min\left\{\bar{\eta}, \sqrt{\frac{4D}{MT}}\right\}$$

in (29) yields

$$\frac{1}{KT}\sum_{k=1}^{K}\sum_{t=0}^{T-1}\mathbb{E}\big[\|\nabla\mathcal{L}_k(\Theta_{k,t})\|_F^2\big] \leq \frac{2}{c_A + c_B}\sqrt{\frac{DM}{T}}, \tag{30}$$

i.e., an $\mathcal{O}(T^{-1/2})$ convergence rate to a stationary point in the smooth non-convex setting, which matches the statement of Theorem 3.1. This completes the proof.

