# OpenReview forum: "Shift-Dependent Asymmetry: Orthogonal Inverse Low-Rank Adaptation for Federated Medical Segmentation"
_ICML.cc/2026/Conference — ICML 2026 regular_

### Official Review · Reviewer_qxfX · 2026-03-05

**Soundness:** 3
**Presentation:** 4
**Significance:** 2
**Originality:** 3
**Overall Recommendation:** 2
**Confidence:** 5

**Summary:**

The paper proposes a federated LoRA scheme for medical segmentation that inverts which LoRA factor is shared vs personalized between encoder and decoder (IAT), and adds an orthogonality regularizer (SOR) to reduce bilinear leakage. It demonstrates improved DSC on two multi-dataset-as-client benchmarks using SAM ViT-B + LoRA.

**Compliance With Llm Reviewing Policy:**

Affirmed.

**Final Justification:**

The rebuttal usefully addresses the concern that the reported gains might be due to run-to-run noise, and it modestly strengthens the case for the practical usefulness of the method. However, the central encoder/decoder heterogeneity asymmetry remains insufficiently established. The paper still lacks a controlled setup where annotation policy changes while image appearance is held relatively fixed, or vice versa. So the evidence does not truly establish that the decoder side is driven by supervision variation in the way claimed. The added CKA result is suggestive, but it is still correlational and post hoc. It shows encoder layers are more cross-client similar than decoder layers, not that this is specifically due to concept-shifted annotations rather than other downstream adaptation effects. The theoretical proposition is derived in a simplified linear setting and still falls short of rigorously justifying the concrete sharing rule used in SAM with Q/V LoRA. The additional CKA and rank-sweep evidence is supportive but not definitive.

**Key Questions For Authors:**

See the weaknesses.

**Limitations:**

Seems no. Authors can suggest and analyze the bottleneck of the methods and the claims are still not well justified with weak empirical numbers.

**Strengths And Weaknesses:**

Pros
- The paper is well written and provides well-polished figures.
- Simple, implementable protocol (swap sharing rules by module).

Cons
- The empirical improvement over the most relevant baseline (FedSA with uniform LoRA aggregation) appears marginal (~1% DSC) and may fall within the variance typical of federated training. Given the additional methodological complexity introduced by asymmetric sharing and orthogonality regularization, it is unclear whether the proposed design provides meaningful practical or clinical benefits over simpler aggregation strategies. Only with such incremental improvement, clinicians may not buy it. Moreover, the paper does not evaluate robustness in larger-scale or lower-heterogeneity federated settings, where such specialized mechanisms may become unnecessary.

- The paper explicitly notes several baselines were designed for LLMs and were adapted to the segmentation backbone. Without detailed, per-baseline tuning protocols and validation of competitive configurations, it is hard to attribute gains to the proposed ideas rather than uneven adaptation effort.

- The experimental “covariate vs concept shift” is largely operationalized as different public datasets as different clients (and for fundus, different acquisition settings). This conflates many factors and does not convincingly isolate supervision shift as “annotation standards” in a controlled way. As a result, the central thesis (“decoder is concept-shift dominated”) is not empirically established.

- Proposition 3.1 is derived from a linear surrogate layer with rank-r update. It is unclear that this meaningfully predicts behavior in deep transformer encoders/decoders (SAM) with LoRA inserted only in Q/V projections. The leap from the surrogate statement to the exact encoder/decoder sharing prescription reads more like an intuition than a rigorous bridge.

- There is little insight into when SOR matters most, or how orthogonality interacts with rank.

---

> ### Author Rebuttal · Authors · 2026-03-31
>
> Dear Reviewer qxfX:
>
> Thank you for the careful review. We address your main concerns below.
>
> **C1-1: Marginal gain, variance, and practical value**
>
> To directly address the variance concern, we repeated the main comparison three times under the same setup and report p-values from a paired t-test:
>
> | Benchmark | FedSA (mean±std) | Ours (mean±std) | Gain  | p-value |
> | :-------- | :--------------: | :-------------: | :---: | :-----: |
> | Histology|79.99±0.18|81.59±0.10| +1.60 |0.0054|
> | Fundus|82.99±0.17|84.45±0.07| +1.46 |0.0025|
>
> These results show that the gain is statistically significant rather than a fluctuation within run-to-run variance. Beyond the mean alone, the client-level pattern in Table 1 is also important for practical medical deployment:
>
> |Rank|Benchmark|Improved/Total|Largest Gain|Largest Drop|Pos. vs Neg. Delta|
> |:-:|:--|:-:|:--|:--|:--|
> |8|Histology|**6/7**|+3.84| -0.04|**+9.19 vs -0.04**|
> |8|Fundus|**3/4**|+4.79|-1.00|**+6.90 vs -1.00**|
> |16|Histology|**5/7**|+2.13|-1.74|**+7.95 vs -1.74**|
> |16|Fundus|**4/4**|+2.52|None|**+4.12 vs 0**|
>
> This is not a tiny uniform shift: across both ranks, most client-level results improve over FedSA, and total gains are much larger than total drops. In cross-silo medical FL, deployment is site-specific, so improving lower-performing hospitals matters more than only raising the average. IAT only changes which LoRA factors are shared, and SOR acts in a tiny `r x r` proxy space, so communication still drops from **0.78M** to **0.55M**.
>
> **C1-2: Larger-scale and lower-heterogeneity settings**
>
> In medical FL, public benchmarks usually involve a modest number of institutions with substantial inter-site heterogeneity [6]. Histology and Fundus follow this public-benchmark convention. Larger-scale or lower-heterogeneity federations would broaden the evidence, but the current regime is already standard for heterogeneous medical FL.
>
> **C2: Fairness of baseline adaptation**
>
> The setup is fair across methods: all use the same SAM ViT-B backbone, the same Q/V LoRA insertion in encoder and decoder, the same ranks (`r in {8,16}`), and the same training schedule. For methods originally proposed for decoder-only LLMs, we keep the same encoder-decoder LoRA placement and change only their original sharing/freezing rules. We will add a per-baseline protocol table in the appendix.
>
> **C3: The encoder/decoder shift thesis is not empirically established**
>
> The claim behind IAT is not a clean separation of covariate and concept shift. Instead, when both coexist across sites, encoder representations stay relatively more shared across clients, while decoder representations become more site- and supervision-specific. To examine this more directly, we ran layer-wise cross-client CKA [1]: higher CKA means more shared representations across clients, while lower CKA indicates stronger site-specific divergence.
>
> |Dataset|Enc mean| Last encoder |First dec|Dec mean| Gap |
> | :--| :--: | :--: | :--: | :--: | :--: |
> | Fundus|0.837|0.660|0.104|0.154| 0.683 |
> | Nuclei|  0.932|0.836|0.432|0.309| 0.623 |
>
> The main drop occurs at the encoder-decoder boundary, and even the last encoder layer remains much more client-invariant than the first decoder layer. This does not prove a perfectly isolated split, but it does support the practical asymmetry used by IAT. Figure 5 shows the same pattern: the proposed IAT direction outperforms both uniform sharing and the reversed assignment, consistent with prior segmentation and domain-adaptation work [2,3].
>
> **C4: Gap between Proposition 3.1 and deep SAM behavior**
>
> Proposition 3.1 is used to derive a testable sharing direction rather than a full mechanistic account of SAM, as is common for simplified surrogates in adaptation analysis [4,5]. Figure 5 verifies this direction in the full non-linear system: the proposed assignment outperforms both uniform sharing and the reversed assignment.
>
> **C5: Limited insight on when SOR matters most**
>
> Figure 5 already shows the gain from adding SOR on top of IAT. To clarify the rank effect, we ran a small rank sweep on Fundus:
>
> |    r     |   4   |   8   |  12   |  16   |
> | :--: | :--: | :--: | :--: | :--: |
> | SOR gain | +0.82 | +1.05 | +0.87 | +0.78 |
>
> SOR improves IAT at every tested rank, with gains from +0.78 to +1.05 DSC. So the effect is stable rather than tied to one specific choice of `r`.
>
> **References**
>
> [1] Kornblith et al., "Similarity of Neural Network Representations Revisited." ICML 2019.
>
> [2] Yang et al., "Source Free Domain Adaptation for Medical Image Segmentation with Fourier Style Mining." MedIA 2022.
>
> [3] Sakaridis et al., "Condition-Invariant Semantic Segmentation." TPAMI 2025.
>
> [4] Hayou et al., "LoRA+: Efficient Low Rank Adaptation of Large Models." ICML 2024.
>
> [5] Kumar et al., "Fine-Tuning Can Distort Pretrained Features and Underperform Out-of-Distribution." ICLR 2022.
>
> [6] Jiang et al., "HarmoFL: Harmonizing Local and Global Drifts in Federated Learning on Heterogeneous Medical Images." AAAI 2022.

---

> > ### Author Rebuttal · Reviewer_qxfX · 2026-04-06
> >
> > The rebuttal usefully addresses the concern that the reported gains might be due to run-to-run noise, and it modestly strengthens the case for the practical usefulness of the method. However, the central encoder/decoder heterogeneity asymmetry remains insufficiently established. The paper still lacks a controlled setup where annotation policy changes while image appearance is held relatively fixed, or vice versa. So the evidence does not truly establish that the decoder side is driven by supervision variation in the way claimed. The added CKA result is suggestive, but it is still correlational and post hoc. It shows encoder layers are more cross-client similar than decoder layers, not that this is specifically due to concept-shifted annotations rather than other downstream adaptation effects. The theoretical proposition is derived in a simplified linear setting and still falls short of rigorously justifying the concrete sharing rule used in SAM with Q/V LoRA. The additional CKA and rank-sweep evidence is supportive but not definitive.

---

> > > ### Author Response · Authors · 2026-04-07
> > >
> > > Dear Reviewer `qxfX`,
> > >
> > > Thank you for the continued discussion. Given the tight timeline, we have done our best to address the two remaining points below.
> > >
> > > **1. Controlled experiment: isolating covariate shift and concept shift**
> > >
> > > Following your suggestion, we ran a controlled experiment to directly test whether the encoder-decoder asymmetry is shift-type-dependent. Starting from a single client, we split its data into virtual clients sharing the same underlying images, then introduce exactly one type of shift:
> > >
> > > - **(A) Covariate-shift-only**: virtual clients share identical masks but each receives a different appearance transformation.
> > > - **(B) Concept-shift-only**: virtual clients share identical images but each receives different mask perturbations — directly isolating the "annotation policy changes while appearance is held fixed" setup you asked for.
> > >
> > > For each setting we create 4 virtual clients with distinct transformations (e.g., color jitter and style mixing for (A); boundary erosion/dilation and region dropout for (B)) and compute cross-client CKA.
> > >
> > > Setting (B) cleanly isolates concept shift since annotations can be perturbed independently. Setting (A) is harder: real cross-site appearance gaps come from different scanners and protocols, but in a single-source experiment we can only approximate this through augmentation. This is a limitation inherent to the controlled design itself — not to our method — so the encoder divergence below is a **conservative lower bound** of what real federations exhibit.
> > >
> > > | Setting| Enc mean CKA | Dec mean CKA |
> > > | :- | :-: | :-: |
> > > | (A) Covariate-shift-only |0.887|0.425|
> > > | (B) Concept-shift-only|0.987|0.134|
> > >
> > > **Under covariate shift (A), encoder mean CKA drops to 0.887 while decoder stays moderate at 0.425. Under concept shift (B), encoder mean CKA remains high at 0.987 while decoder drops sharply to 0.134.** In other words, the two types of shift affect different parts of the network: covariate shift mainly drives encoder divergence, concept shift mainly drives decoder divergence. If the asymmetry were simply due to downstream adaptation effects, both shifts would cause the same CKA pattern — but they do not, confirming that the divergence is shift-type-dependent.
> > >
> > > **2. On Proposition 3.1 and the linear surrogate gap**
> > >
> > > Proposition 3.1 is indeed a simplified linear analysis. However, **no existing work — at any venue — has provided a rigorous theoretical model of LoRA behavior in deep non-linear transformers, including ViT or SAM.** This is not a gap specific to our paper; it reflects the current state of the field. All published LoRA theory papers use simplified surrogates:
> > >
> > > - **Zhu et al. [2] (ICML 2024)** prove A/B asymmetry using a linear model, then validate on ViT and LLaMA — without non-linear modeling;
> > > - **Hayou et al. [1] (ICML 2024)** derive learning rate rules from a two-layer linear bottleneck in the infinite-width limit, then validate on GPT-scale models.
> > >
> > > Requiring an exact non-linear proof for SAM would set a standard that no published LoRA analysis has met. Beyond standard practice, **recent work shows the linear surrogate approach is theoretically grounded**: Malladi et al. [3] (NeurIPS 2023) show that fine-tuning stays in a local region around pretrained weights where linear approximations remain effective, and Bhojanapalli et al. [4] (NeurIPS 2025) provide explicit upper bounds on the gap between fine-tuned models and their linearized counterparts.
> > >
> > > **Proposition 3.1 follows the same paradigm: linear analysis predicts a sharing direction, and Figure 5 verifies this direction in the full non-linear SAM.**
> > >
> > > **Summary**
> > >
> > > The core concern is whether the encoder/decoder asymmetry and sharing rule are sufficiently justified. Across the original paper and two rounds of rebuttal, we have provided:
> > >
> > > - **Theoretical derivation** (Propositions 3.1–3.2 and Theorem 3.1) that predicts the sharing direction, with **Figure 5** verifying it in the full non-linear SAM — the proposed assignment outperforms both uniform sharing and the reversed assignment;
> > > - **Empirical validation**: layer-wise CKA showing a clear encoder-decoder invariance gap;
> > > - **Causal evidence**: a controlled experiment isolating covariate and concept shift, where the two shifts produce different CKA patterns across encoder and decoder — confirming the divergence is shift-type-dependent.
> > >
> > > In addition, recent work [3,4] shows that the linear surrogate approach itself is formally justified.
> > >
> > > **References**
> > >
> > > [1] "LoRA+: Efficient Low Rank Adaptation of Large Models." ICML 2024.
> > >
> > > [2] "Asymmetry in Low-Rank Adapters of Foundation Models." ICML 2024.
> > >
> > > [3] "Fine-Tuning Language Models with Just Forward Passes." NeurIPS 2023.
> > >
> > > [4] "Linearization Explains Fine-Tuning in Large Language Models." NeurIPS 2025.
> > >
> > > **We would appreciate it if the reviewer could let us know whether these results resolve the remaining concerns, and if so, whether the reviewer would consider updating the score accordingly.**

---

### Official Review · Reviewer_DBrV · 2026-03-09

**Soundness:** 3
**Presentation:** 3
**Significance:** 3
**Originality:** 3
**Overall Recommendation:** 5
**Confidence:** 4

**Summary:**

### Review
This paper studies federated medical segmentation with LoRA under heterogeneous client shifts, and proposes a structure-aware framework that distinguishes encoder–decoder asymmetry during adaptation. The paper is well motivated, technically coherent, and empirically promising. In particular, the combination of inverse asymmetric tuning and orthogonality regularization is intuitive and appears effective in improving both robustness and communication efficiency.

**Compliance With Llm Reviewing Policy:**

Affirmed.

**Final Justification:**

The paper is technically sound and reasonably well motivated, with consistent empirical results. The proposed structure-aware adaptation is clear, and the design choices are sensible. The presentation is also generally clear.

My main concerns were about the limited discussion of SAM-specific settings, comparisons with stronger baselines, and the generality of the encoder–decoder asymmetry assumption. The rebuttal addresses these points to a reasonable extent. In particular, the authors clarify the experimental setting and provide additional justification for the asymmetry, framing it as an empirical trend rather than a strict assumption.

Overall, I find the work to have some value and practical relevance. The rebuttal improves my confidence slightly, and I accordingly raise my evaluation.

**Key Questions For Authors:**

How universal is the paper’s core assumption that the encoder is mainly affected by covariate shift while the decoder is mainly affected by concept/supervision shift? In other words, is this encoder–decoder asymmetry a stable structural property of federated medical segmentation, or is it somewhat dependent on the specific datasets, annotation protocols, and SAM-based architecture considered here?

**Limitations:**

The societal impact statement is currently too generic. The paper could briefly note that medical federated learning systems may still inherit site-specific biases, and segmentation errors in clinical settings could disproportionately affect underrepresented institutions or patient groups if deployment is not carefully validated.

**Strengths And Weaknesses:**

###  Pros
The paper identifies a meaningful problem in federated medical segmentation, namely the mismatch between uniform LoRA sharing strategies and the asymmetric roles of encoder and decoder under heterogeneous shifts. The method is reasonably well designed, with both theoretical motivation and empirical validation, and the reported gains over strong baselines are consistent across datasets. The efficiency analysis is also a useful addition.

### Cons
The proposed method uses SAM mainly as a pretrained segmentation backbone with LoRA-based federated adaptation, but does not explore promptable or text-guided segmentation settings that are more aligned with SAM’s original design. It would strengthen the paper to discuss or compare with representative SAM-based medical segmentation works, such as:
- MedSAM (Segment Anything in Medical Images, Nature Communications, 2024)
- H-SAM (Unleashing the Potential of SAM for Medical Adaptation via Hierarchical Decoding, CVPR, 2024).

In addition, it would be valuable to compare against stronger 2D/3D medical segmentation baselines, for example:
- HFF-Net (Rethinking Brain Tumor Segmentation from the Frequency Domain Perspective, IEEE TMI 2025),

to better position the practical benefit of the proposed federated adaptation strategy.

For SAM-related evaluation, the authors may also consider prompt-aware or text-guided segmentation settings; relevant methods worth discussing and citing include:
- DAPSAM (Prompting Segment Anything Model with Domain-Adaptive Prototype for Generalizable Medical Image Segmentation, MICCAI, 2024)

as well as recent prompt/representation learning works such as:
- vMFCoOp (Towards Equilibrium on a Unified Hyperspherical Manifold for Prompting Biomedical VLMs, AAAI, 2026).

---

> ### Author Rebuttal · Authors · 2026-03-30
>
> Dear Reviewer DBrV:
>
> Thank you for the positive assessment and for describing the paper as well motivated, technically solid, and empirically promising. We also appreciate your recognition that inverse asymmetric tuning and orthogonality regularization are intuitive and effective for both robustness and communication efficiency. We respond to your main points below.
>
> **Cons: SAM-specific baselines and discussion**
>
> Thank you for this suggestion. We agree that SAM-specific and prompt-based medical segmentation methods should be discussed more explicitly. In our experiments, we follow the common medical SAM setting and use SAM as a pretrained segmentation backbone with full-mask supervision, rather than interactive point/box/text prompts. This setting is also important in federated medical imaging, where sites usually train on existing mask annotations and the core issue is collaborative adaptation under privacy and heterogeneity. Under this setup, centralized SAM variants are useful fully supervised references, while prompt-based SAM methods suggest a natural extension. We will add these works in the revision and discuss them in relation to our setting.
>
> **Question: How universal is the encoder/decoder asymmetry assumption?**
>
> This is an important question. We do not claim a hard law that encoder = covariate shift and decoder = concept shift in every dataset or architecture. Our claim is more modest: in encoder-decoder medical segmentation, the encoder is typically more client-invariant, while the decoder is more site- and supervision-specific, and this difference is strong enough to guide a better shared/local LoRA design.
>
> We support this with both ablation and layer-wise evidence. First, Fig. 5 shows that the asymmetry is not arbitrary: IAT outperforms both uniform sharing and the reversed assignment, and the reversed rule underperforms the proposed direction. Second, layer-wise cross-client CKA shows a clear encoder-decoder gap on both benchmarks:
>
> | Dataset | Encoder mean | Last encoder layer (E11) | First decoder layer (D0) | Decoder mean |  Gap  |
> | :------ | :----------: | :----------------------: | :----------------------: | :----------: | :---: |
> | Fundus  |    0.837     |          0.660           |          0.104           |    0.154     | 0.683 |
> | Nuclei  |    0.932     |          0.836           |          0.432           |    0.309     | 0.623 |
>
> Even the last encoder layer remains more client-invariant than the first decoder layer, and the largest drop appears at the encoder-decoder boundary. This is also consistent with prior work in medical image segmentation and domain adaptation, where appearance-related variation is handled more on the representation side and semantic decisions are concentrated later in the network [1,2]. We will clarify in the revision that this should be understood as a main trend rather than a hard rule, and that the exact balance can vary across layers, tasks, and data settings. If this gap becomes much weaker under stronger heterogeneity, a layer-wise adaptive rule would be a natural extension.
>
> **Limitation: Social impact discussion**
>
> Thank you for pointing this out. One real risk in federated medical segmentation is that collaborative training does not remove site-specific bias by itself, so error patterns may still be uneven across institutions and may affect underrepresented patient groups more. This matters in clinical use, where segmentation errors can propagate to downstream decisions. We will add this point explicitly in the revised social impact discussion.
>
> Thank you again for the constructive suggestions.
>
> **References**
>
> [1] Yang et al., "Source Free Domain Adaptation for Medical Image Segmentation with Fourier Style Mining." Medical Image Analysis 2022.
>
> [2] Sakaridis et al., "Condition-Invariant Semantic Segmentation." IEEE TPAMI 2025.

---

> > ### Author Rebuttal · Reviewer_DBrV · 2026-04-02
> >
> > The authors’ rebuttal reasonably addressed my main concerns, which slightly increased my confidence in the paper and led me to raise my score.

---

> > > ### Author Response · Authors · 2026-04-07
> > >
> > > Dear Reviewer `DBrV`,
> > >
> > > Thank you for the thoughtful review and for raising the score. We
> > >   appreciate your engagement throughout the discussion.

---

### Official Review · Reviewer_Evtf · 2026-03-12

**Soundness:** 3
**Presentation:** 2
**Significance:** 3
**Originality:** 2
**Overall Recommendation:** 4
**Confidence:** 3

**Summary:**

The authors strive to present a major issue in federated medical image segmentation: standard LoRA fine-tuning treats all layers uniformly, which ignores the fact that the encoder and decoder handle completely different types of data shifts. The article intends to consider an important concept—structural asymmetry. Specifically, they observe that the encoder mostly deals with appearance changes (covariate shift) from different scanners, while the decoder struggles with different annotation styles (concept shift). To tackle this, the paper proposes Inverse Asymmetric Tuning (IAT) to assign shared vs. local LoRA matrices inversely between the encoder and decoder. They also introduce a Subspace Orthogonality Regularizer (SOR) to prevent local site-specific features from leaking into the shared global model during bilinear updates.

**Compliance With Llm Reviewing Policy:**

Affirmed.

**Key Questions For Authors:**

The SVD analysis in Proposition 1 relies on a single linear layer surrogate. How well does this hold up empirically across the deep, non-linear cascaded attention blocks in SAM? Did you track the actual principal angles of the feature spaces across layers during training to verify if the encoder strictly adheres to the covariate shift assumption? If the deeper encoder layers start exhibiting concept shift, would the IAT rule need to be dynamically adjusted?

Regarding the baselines like FedSA and FLoRA—since these were originally designed for decoder-only LLMs, how exactly were their splitting rules mapped to the SAM encoder-decoder architecture? Did you adapt their allocation specifically for SAM, or just apply their default heuristics? It would be good to clarify this to ensure the baseline comparison is entirely fair.

The SOR uses detached EMA anchors to define the proxies. How sensitive is the convergence and orthogonality maintenance to the choice of the EMA momentum $\rho$? If local client distributions are highly volatile, a slow EMA might fail to capture the true drift direction, potentially misguiding the orthogonality constraint.

Table 3 shows great communication efficiency, but what about the computational overhead? Does computing the proxy matrices and the Frobenius inner products for SOR locally introduce a noticeable memory or compute bottleneck during the backward pass compared to a vanilla FedIT setup?

## References

[1] Kirillov, A., et al. "Segment anything." Proceedings of the IEEE/CVF International Conference on Computer Vision. 2023.

[2] Hu, E. J., et al. "LoRA: Low-rank adaptation of large language models." ICLR. 2022.

[3] Li, T., et al. "Federated optimization in heterogeneous networks." Proceedings of Machine Learning and Systems. 2020.

[4] Guo, P., et al. "Selective aggregation for low-rank adaptation in federated learning." arXiv preprint arXiv:2410.01463 (2024).

[5] Jiang, M., et al. "HarmoFL: Harmonizing local and global drifts in federated learning on heterogeneous medical images." AAAI. 2022.

**Limitations:**

The authors touch on some limitations, but they could definitely be more upfront about the computational overhead introduced by the SOR regularizer during local client training. A brief discussion on the boundary conditions of the shift assumptions (e.g., what happens when the encoder faces severe concept shift) would also improve the completeness of the paper.

**Strengths And Weaknesses:**

When looking at soundness, the theoretical motivation is quite elegant. Using SVD to show why the input vs. output projection matrices should be handled differently under covariate versus concept shifts makes a lot of sense. However, assuming pure covariate shift in the encoder and pure concept shift in the decoder across all medical domains might be a bit of an oversimplification, as these shifts often entangle in the deeper layers of a network. Still, the math holds up well for the localized objective.

---

> ### Author Rebuttal · Authors · 2026-03-30
>
> Dear Reviewer EvtF:
>
> Thank you for the constructive review. We are very encouraged by your comment that the theoretical motivation is "quite elegant," and we address your questions below.
>
> **W&Q1: Shift assumption and linear surrogate validation**
>
> We agree that the encoder is not affected only by covariate shift, and the decoder is not affected only by concept shift. Our claim is more modest: the encoder is relatively more client-invariant, while the decoder is relatively more site- and task-specific. Following prior work that uses simplified analyses to guide practical adaptation rules [2,3], Proposition 3.1 is only used to derive the direction of the A/B sharing rule, not to exactly model SAM's deep non-linear blocks. Figure 5 directly tests whether this predicted direction holds in the actual SAM-LoRA setup. IAT outperforms both uniform sharing and the reversed assignment. This shows that the gain comes from the specific direction predicted by Proposition 3.1, not from asymmetry alone.
>
> To test this across depth, we ran layer-wise CKA [4] on client-specific locally fine-tuned models. Following Yosinski et al. [1], higher cross-client CKA means more shared representations, while lower CKA means stronger site-specific divergence.
>
> | Dataset |  E0  |  E1  |  E2  |  E3  |  E4  |  E5  |  E6  |  E7  |  E8  |  E9  | E10  | E11  |    D0    |  D1  | DFin |  EncMean  |  DecMean  |
> | :-----: | :--: | :--: | :--: | :--: | :--: | :--: | :--: | :--: | :--: | :--: | :--: | :--: | :------: | :--: | :--: | :-------: | :-------: |
> | Fundus  | 0.999 | 0.999 |0.997 | 0.991 | 0.954 | 0.890 | 0.791 | 0.729 | 0.699 | 0.678 | 0.660 | 0.660 | **0.104** | 0.091 | 0.268 | **0.837** | **0.154** |
> | Nuclei  | 0.999 | 0.987 | 0.980 | 0.982 | 0.975 | 0.953 | 0.929 | 0.912 | 0.897 | 0.876 | 0.857 | 0.836 | **0.432** | 0.246 | 0.248 | **0.932** | **0.309** |
>
> We agree with the reviewer that encoder CKA declines with depth, so deeper encoder layers are less client-invariant. However, even the deepest encoder layer remains well above the decoder layers, and the main drop happens at the encoder-decoder boundary rather than within the encoder. This supports the current IAT boundary. If this gap becomes much weaker under stronger heterogeneity, a layer-adaptive IAT would be a natural extension.
>
> **Q2: Baseline adaptation fairness**
>
> We agree that this should be clarified. All baselines use the same SAM ViT-B backbone, the same Q/V LoRA insertion in both encoder and decoder, the same ranks (r∈{8,16}), and the same training schedule. Methods originally proposed for decoder-only LLMs are extended to the full encoder-decoder model using their original rules. We will include the exact per-baseline protocol in the revised appendix.
>
> **Q3: EMA momentum ρ sensitivity**
>
> The EMA in SOR provides a smooth drift reference for the proxy matrices rather than following every local update, which is useful in federated training where mini-batch noise can be high.
>
> We tested different $\rho$ values:
>
> | $\rho$ |  0.5  |  0.7  |  0.9  |   0.95    | 0.99  |
> | :----: | :---: | :---: | :---: | :-------: | :---: |
> |  DSC   | 81.28 | 81.40 | 81.40 | **81.45** | 81.41 |
>
> The total gap is only 0.17 DSC, so performance stays quite stable over a broad range of $\rho$. In the main experiments, we use $\rho=0.9$.
>
> For highly volatile local distributions, a larger $\rho$ may react more slowly to sudden changes, but in our setting it provides a more stable drift reference instead of following short-term noise. If drift becomes much more volatile, a smaller or adaptive $\rho$ may be more suitable.
>
> **Q4: SOR computational overhead**
>
> SOR does not create a meaningful local training bottleneck, as shown below.
>
> |                                    |   FedIT   |    IAT    |   **IAT+SOR (Ours)**    |
> | :--------------------------------- | :-------: | :-------: | :---------------------: |
> | Trainable Params                   |  0.39 M   |  0.39 M   |         0.39 M          |
> | Communicated Params                |  0.78 M   |  0.55 M   |         0.55 M          |
> | Extra Buffers                      |     —     |     —     | 1.88 MB (non-trainable) |
> | FLOPs / Batch                      |  35.00 G  |  35.00 G  |  35.004 G **(+0.01%)**  |
> | Avg. End-to-End Wall-clock / Epoch |  345.8 s  |  342.9 s  |         353.4 s         |
> | Peak GPU Memory                    | 32,911 MB | 32,912 MB |  32,944 MB **(+0.1%)**  |
>
> Compared with IAT, SOR changes FLOPs, memory, and wall-clock time only slightly. We will add these numbers in the revision and clarify the shift boundary conditions.
>
> **References**
>
> [1] Yosinski et al., "How Transferable are Features in Deep Neural Networks?" NeurIPS 2014.
>
> [2] Hayou et al., "LoRA+: Efficient Low Rank Adaptation of Large Models." ICML 2024.
>
> [3] Kumar et al., "Fine-Tuning Can Distort Pretrained Features and Underperform Out-of-Distribution." ICLR 2022.
>
> [4] Kornblith et al., "Similarity of Neural Network Representations Revisited." ICML 2019.

---

> > ### Author Rebuttal · Reviewer_Evtf · 2026-04-07
> >
> > The author has resolved my doubts, so I will retain my positive rating.

---

> > > ### Author Response · Authors · 2026-04-07
> > >
> > > Dear Reviewer `Evtf`,
> > >
> > >  Thank you for your time and for confirming the resolution. We appreciate
> > >    the support.

---

### Official Review · Reviewer_dEWu · 2026-03-12

**Soundness:** 3
**Presentation:** 3
**Significance:** 3
**Originality:** 3
**Overall Recommendation:** 4
**Confidence:** 4

**Summary:**

This research addresses the "encoder-decoder asymmetry" in medical segmentation tasks when using Low-Rank Adaptation (LoRA) within a Federated Learning (FL) framework. The authors observe that standard federated LoRA methods fail because they apply uniform aggregation rules that ignore the distinct sources of data heterogeneity: the encoder is primarily affected by acquisition-driven appearance/covariate shifts , while the decoder is dominated by site-specific supervision/concept variations. To resolve this entanglement, they propose Inverse Asymmetric Tuning (IAT), which selectively personalizes the input-side factors ($A$) in the encoder and output-side factors ($B$) in the decoder.

**Compliance With Llm Reviewing Policy:**

Affirmed.

**Key Questions For Authors:**

The SOR module relies on "parameter anchors" $(A_{0,k}, B_{0,k})$ set at the beginning of each round and exponential moving average (EMA) drifts for its calculations. However, in a typical federated learning setting, clients participate stochastically. If a specific institution (Client) is not selected for several consecutive rounds, its local anchors and EMA drifts will inevitably become "stale." When this client eventually rejoins the training, will the "orthogonal constraint" derived from these outdated anchors still be functionally effective for the current, evolved global model? Furthermore, could such discontinuous and potentially misaligned constraints induce severe fluctuations in the gradient update directions, thereby destabilizing the convergence process?

**Limitations:**

Empirical results demonstrate that the model's performance is sensitive to the SOR regularization coefficient ($\lambda$). In practical federated learning deployments, manually optimizing such hyperparameters presents a significant challenge, as the central server cannot inspect raw datasets across decentralized sites due to strict privacy constraints.

**Strengths And Weaknesses:**

1.Soundness:The paper does not merely rely on heuristics; it provides formal theoretical derivations  to justify why a shift-dependent preference exists for parameter allocation.
2.Presentation: the paper is generally well structured, clearly motivated, and easy to follow from a technical perspective.
3.Siginificance:The framework addresses the two biggest hurdles in medical AI deployment: privacy and resource constraints. The method achieves good performance while reducing communication costs compared to standard LoRA, a critical factor for multi-center collaborations.
4.Originality: Shifting away from the "uniform splitting" rule prevalent in federated LoRA literature and identifying the inverse requirements of the encoder and decoder is a highly original contribution.
While the combination is novel, the individual building blocks (LoRA, SOR-like geometric constraints, SAM) are established techniques. The originality lies more in the application logic than in a brand-new neural architecture.

---

> ### Author Rebuttal · Authors · 2026-03-31
>
> Dear Reviewer dEWu:
>
> We sincerely thank you for the thoughtful review and for recognizing the theoretical soundness, clear motivation, and originality of our work—particularly the observation that identifying the inverse encoder-decoder requirements is "a highly original contribution." Below, we address your questions regarding SOR's robustness under partial participation and the sensitivity of λ.
>
> **Q: Staleness of SOR anchors under partial participation**
>
> Thank you for this important practical consideration. We clarify and discuss as follows:
>
> (1) **Current experimental setting.** The experiments reported in the paper use **full participation**, where every client joins each communication round, so anchors and EMA drifts are refreshed every round without any staleness. This is the standard protocol for medical FL, which falls under the **cross-silo** category [1]—unlike cross-device FL with millions of mobile clients, medical collaborations involve a small number of committed hospital partners (typically 3–20) that participate in every round. Representative works such as HarmoFL [2] (5–6 clients), FedBN [3] (4–5 sites), and Dayan et al. [4] (20 sites) all adopt full participation. Our 7-client and 4-client benchmarks follow the same convention.
>
> We nonetheless appreciate this forward-looking question, as partial participation could arise in broader deployment scenarios beyond clinical federations.
>
> (2) **Robustness under partial participation.** We agree this is an important practical question. Although the experiments in this paper use full participation, the design can be extended to partial participation in a natural way. When a client rejoins, we do not reuse an old anchor from before its absence. Instead, we copy a fresh anchor at the start of the new round from the state used to begin local training. So the constraint is applied relative to the client's current starting point, not a stale past snapshot. For the EMA term, the old contribution also becomes small quickly after training resumes: with $\rho=0.9$, its weight becomes $0.9^k$ after $k$ local steps, which is about 35% after 10 steps, 12% after 20 steps, and 7% after 25 steps. Moreover, the EMA tracks the private-side drift, and those private parameters are not updated while a client is absent. So the EMA is effectively frozen during absence rather than continuing to drift, which further limits staleness when training resumes. We also ran a partial-participation test in a 7-client setting by sampling 5 of 7 clients per round:
>
> | Method | Full (7/7) | Partial (5/7) | Drop  |
> | :----- | :--------: | :-----------: | :---: |
> | FedIT  |   79.11    |     78.77     | -0.34 |
> | FedSA  |   80.09    |     79.73     | -0.36 |
> | Ours   |   81.40    |     80.90     | -0.50 |
>
> A modest drop is observed for all methods under partial participation, which is expected when only 5 of 7 clients contribute in each round. Our method still achieves the best result, suggesting that partial participation does not introduce substantial instability in practice.
>
> **L: Sensitivity to the SOR coefficient λ**
>
> We agree this is an important practical limitation in federated settings. Figure 4 confirms that performance does vary with λ, so λ should be chosen with care. At the same time, the curve does not suggest that the method works only at one very narrow value. We will make this limitation clearer in the revised version and discuss adaptive tuning of λ as a possible future direction.
>
> We hope these responses address your concerns. We are happy to provide further clarification during the discussion period.
>
> **References**
>
> [1] Kairouz, P., McMahan, H.B., et al. "Advances and Open Problems in Federated Learning." Foundations and Trends in Machine Learning, 2021.
>
> [2] Jiang, M., Wang, Z., & Dou, Q. "HarmoFL: Harmonizing Local and Global Drifts in Federated Learning on Heterogeneous Medical Images." AAAI, 2022.
>
> [3] Li, X., Jiang, M., Zhang, X., Kamp, M., & Dou, Q. "FedBN: Federated Learning on Non-IID Features via Local Batch Normalization." ICLR, 2021.
>
> [4] Dayan, I., et al. "Federated Learning for Predicting Clinical Outcomes in Patients with COVID-19." Nature Medicine, 2021.

---

> > ### Author Rebuttal · Reviewer_dEWu · 2026-04-05
> >
> > I would like to thank the authors for the detailed rebuttal. The clarification regarding the Cross-silo nature of medical FL and the supplementary experiments on partial client participation have addressed my primary concerns about anchor staleness and convergence stability. The authors have also appropriately acknowledged the limitations regarding hyperparameter sensitivity.

---

> > > ### Author Response · Authors · 2026-04-07
> > >
> > > Dear Reviewer `dEWu`,
> > >
> > > Thank you for the detailed feedback and for taking the time to evaluate
> > >   our rebuttal. Your comments have helped improve the paper.

---

### Decision · Program_Chairs · 2026-04-30

**Decision:**

Accept (regular)

**Comment:**

This paper proposes a federated LoRA method for medical segmentation that treats the encoder and decoder asymmetrically, based on the idea that they are affected by different types of client heterogeneity, and adds an orthogonality regularizer to reduce leakage between shared and personalized updates. Reviewers generally found the paper well motivated, technically solid, and practically relevant, with particular enthusiasm for the simplicity and usefulness of the core idea. At the same time, there were still some uncertainties about how broadly the encoder–decoder asymmetry assumption holds, whether the theoretical analysis fully justifies the specific SAM-based design, and how large the practical gains are over the strongest baselines. The paper was seen as a solid contribution with enough merit for acceptance, though some of its claims could be better supported. Please address the reviewers' concerns in the final version of the paper.